# Simplicity Prevails: Rethinking Negative Preference Optimization for LLM Unlearning

## Abstract

In this work, we study the problem of large language model (LLM) unlearning, aiming to remove unwanted data influences and associated model capabilities (*e.g.*, copyrighted data or harmful content generation) while preserving essential model utilities, without the need for retraining from scratch. Despite the growing need for LLM unlearning, a technically-grounded optimization framework remains lacking. For example, gradient ascent (GA)-type approaches have been widely used for unlearning. However, GA is suboptimal as it reverses the learning process without controlling the optimization divergence speed, making it prone to over-forgetting and even leading to model collapse. Negative preference optimization (NPO) has been proposed to address this issue and is considered one of the state-of-the-art LLM unlearning approaches. In this work, we revisit NPO, and identify the issue of reference model bias, which could undermine NPO's effectiveness, particularly when unlearning forget data of varying difficulty. This reference model bias arises from using the reference model as the basis for unlearning criterion in NPO, leading to two issues: uneven optimization power allocation over the forget data and ineffective gradient weight smoothing in the early optimization stages. To overcome these challenges, we propose a simple yet effective unlearning optimization framework, called SimNPO, showing that 'simplicity' in removing the reliance on a reference model (through the lens of simple preference optimization) benefits unlearning. We also provide deeper insights into SimNPO's advantages, supported by analysis using mixtures of Markov chains. Furthermore, we present extensive experiments validating SimNPO's superiority over existing baselines in benchmarks like TOFU and MUSE, and robustness against relearning attacks.

## 1 Introduction

The rapid advancement of large language models (LLMs) has raised security and safety concerns, including issues related to copyright violations and sociotechnical harms (Huang et al., 2024; Wang et al., 2023; Li et al., 2024; Shi et al., 2024). However, retraining these models to remove undesirable data influences is often impractical due to the substantial costs and time required for such processes. This gives rise to the problem of **LLM unlearning**, which aims to effectively remove undesired data influences and/or model behaviors while preserving the utility for essential, unrelated knowledge generation, and maintaining efficiency without the need for retraining (Eldan & Russinovich, 2023; Yao et al., 2023; Liu et al., 2024b; Blanco-Justicia et al., 2024).

To trace its origins, the concept of *machine unlearning* was initially developed for data removal to comply with privacy regulations such as the "right to be forgotten" (Rosen, 2011; Hoofnagle et al., 2019), with early studies focusing on vision models (Cao & Yang, 2015; Warnecke et al., 2021; Bourtoule et al., 2021; Thudi et al., 2022; Kurmanji et al., 2024; Jia et al., 2023; Gandikota et al., 2023; Fan et al., 2024b). However, it is soon adapted to LLMs to remove unwanted data, knowledge, or specific model capabilities (Eldan & Russinovich, 2023; Yao et al., 2023; Liu et al., 2024b; Ji et al., 2024; Li et al., 2024; Shi et al., 2024; Maini et al., 2024; Zhang et al., 2024a; Jia et al., 2024). The current optimization foundation for LLM unlearning often relies on *optimization divergence* from the pre-trained state. Divergence refers to the process of deviating from the converged pre-trained model state to reverse the effects of learning the forgotten data, thereby achieving unlearning (Liu et al., 2022a; Maini et al., 2024; Yao et al., 2023; Jia et al., 2024). Nevertheless, the lack of control over optimization divergence can result in either under-forgetting, where insufficient unwanted data

influence is removed, or over-forgetting, leading to a significant loss of model utility in LLMs. Therefore, optimization for LLM unlearning is a highly non-trivial challenge.

Negative preference optimization (**NPO**) (Zhang et al., 2024a) emerges as an effective approach for LLM unlearning, as demonstrated by its better control of the divergence speed during unlearning optimization and its strong performance in current benchmarks such as TOFU (Maini et al., 2024) and MUSE (Shi et al., 2024). Inspired by direct preference optimization (DPO) (Rafailov et al., 2024), it treats the forget data points as negative responses, providing a lower-bounded unlearning objective. This also induces a gradient weight smoothing scheme to regulate the speed of divergence. We refer readers to Sec. 3 for details.

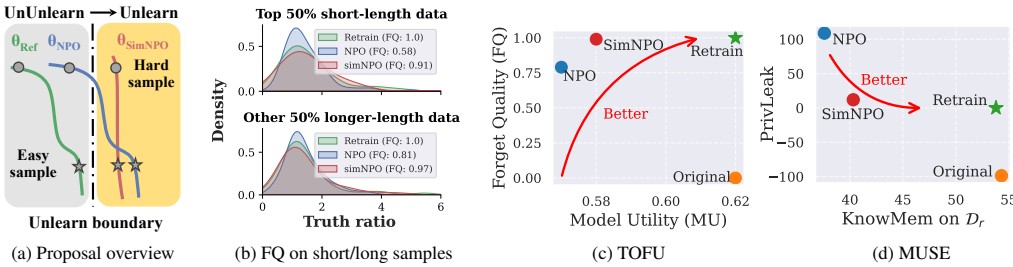

(a) Proposal overview     (b) FQ on short/long samples     (c) TOFU     (d) MUSE

Figure 1: *(a)* Systematic overview of an LLM ($\theta$) post-unlearning using the proposed SimNPO, compared to NPO (Zhang et al., 2024a) and the reference model, where NPO struggles to unlearn hard samples due to the reference model bias. *(b)* Truth ratio distribution of top 50% shortest-length forget data points and the other 50% longer-length data for NPO, SimNPO, and Retrain on the TOFU Forget05 dataset (Maini et al., 2024), where SimNPO achieves better forget quality (FQ) than NPO and exhibits a truth ratio distribution closer to Retrain. Noet that FQ is a statistical measure quantifying the closeness between the truth ratio distribution of an unlearned model and that of Retrain (with FQ= 1 representing optimal unlearning). *(c) & (d)* Experiment highlights on TOFU Forget05 and MUSE News datasets (Shi et al., 2024). Unlearning effectiveness is measured by FQ for TOFU and PrivLeak for MUSE, while utility preservation is evaluated using model utility for TOFU and KnowMem on retain data for MUSE (see Table 1). In both tasks, Retrain is the gold standard for unlearning.

Despite the advancements NPO has introduced to the optimization foundation for LLM unlearning, this work will identify its potential limitations for the first time, arising from its reliance on the reference model (*i.e.*, the model prior to unlearning) as the basis for promoting and regulating the optimization divergence. We refer to this issue as *reference model bias*. **Fig. 1-(a)** illustrates this issue schematically. NPO aims to widen the gap between the unlearned model ($\theta_{\text{NPO}}$) and the reference model ($\theta_{\text{ref}}$). However, the prediction confidence of $\theta_{\text{ref}}$ varies across samples (green line in Fig. 1-(a)). For some samples, predictions of $\theta_{\text{ref}}$ are already near the unlearning boundary, making them "easy examples" for unlearning, where further enlarging the gap is unnecessary. Despite this, NPO may continue increasing the distance (blue line in Fig. 1-(a)), causing easy examples to move far beyond the boundary while "hard examples" remain unresolved. Throughout this work, the key research question we aim to answer is:

> *(Q) When and why does the current optimization foundation –NPO–for LLM unlearning become ineffective, and how can it be improved?*

To address **(Q)**, we propose a simple yet effective unlearning optimization framework, termed **SimNPO**, demonstrating that properly removing reliance on a reference model can significantly enhance unlearning. This approach also draws inspiration from simple preference optimization in LLM alignment (Meng et al., 2024). Additionally, we provide detailed and in-depth insights into how SimNPO overcomes the limitations of NPO caused by reference model bias. For example, Fig. 1-(a) schematically illustrates that, compared to NPO, SimNPO more accurately recognizes the unlearning data difficulty and provides better optimization allocation across different types of forget data, *e.g.*, hard vs. easy samples in Fig. 1-(a). **Fig. 1-(b)** further provides experimental justification by comparing the unlearning performance of NPO and SimNPO across forget data points, categorized by *response length*. The rationale is that as noted in (Meng et al., 2024), the reference model tends to bias toward generating longer but lower-quality sequences, making these longer samples easier to unlearn. However, NPO may exacerbate this by over-allocating optimization power to these easy samples, thereby disadvantaging the unlearning of shorter-response forget data. This explains why Fig. 1-(b) shows that NPO performs worse than SimNPO, as evidenced by a greater deviation from **Retrain** (the exact unlearning method achieved by retraining the model from scratch without the

forgotten data). This gap is particularly evident when unlearning the top 50% shortest-length forget data compared to the longer 50% of the forget set. We refer readers to Secs. 4 and 5 for more details.

The contributions of our work are summarized below:

• We revisit the NPO framework and identify its potential weakness–reference model bias–in LLM unlearning, which can lead to issues such as sensitivity to the reference model's response quality and ineffective gradient weight smoothing.

• Building on insights into NPO's limitations, we propose an improved LLM unlearning approach, SimNPO, which extends NPO using a reference-free optimization framework, simple preference optimization (Meng et al., 2024). We also delve into the technical rationale behind how SimNPO alleviates the limitations of NPO, validated through the lens of mixtures of Markov chains.

• We conduct extensive experiments to demonstrate the improvements of SimNPO over NPO across various scenarios, including: forgetting data with different response lengths, as in TOFU (Maini et al., 2024); forgetting data with uniform response lengths, as in MUSE (Shi et al., 2024) and WMDP (Li et al., 2024); and defending against relearning-based attacks (Lynch et al., 2024; Hu et al., 2024). See some experiment highlights in **Fig. 1-(c,d)**.

## 2 RELATED WORK

**Machine unlearning.** The gold standard for machine unlearning in our work is 'Retrain', also referred to as *exact* unlearning (Cao & Yang, 2015; Thudi et al., 2022; Fan et al., 2024a), which involves retraining the model from scratch on the training set while excluding the data points to be forgotten. However, exact unlearning is challenging in practice due to the assumption for access to the full training set and the high computational cost of retraining. To address these challenges, various *approximate* unlearning methods have been developed (Nguyen et al., 2022; Bourtoule et al., 2021; Triantafillou et al., 2024). These approaches typically involve model fine-tuning or editing, applied to the pre-trained model, based on the unlearning request. Their effectiveness has been shown in different application domains, including image classification (Liu et al., 2022b; Jia et al., 2023; Kurmanji et al., 2024; Fan et al., 2024a), image generation (Gandikota et al., 2023; Fan et al., 2024b; Zhang et al., 2024b), federated learning (Liu et al., 2022c; Halimi et al., 2022; Jin et al., 2023), and graph neural networks (Chen et al., 2022; Chien et al., 2022; Wu et al., 2023a).

**LLM unlearning.** There has also been a growing body of research focusing on machine unlearning for LLMs (Lu et al., 2022; Jang et al., 2022; Kumar et al., 2022; Zhang et al., 2023; Pawelczyk et al., 2023; Eldan & Russinovich, 2023; Ishibashi & Shimodaira, 2023; Yao et al., 2023; Maini et al., 2024; Zhang et al., 2024a; Li et al., 2024; Wang et al., 2024; Jia et al., 2024; Liu et al., 2024b;a; Thaker et al., 2024; Kadhe et al., 2024). Applications of unlearning in LLMs are diverse, from safeguarding copyrighted and personally identifiable information (Jang et al., 2022; Eldan & Russinovich, 2023; Wu et al., 2023b), to preventing LLMs from creating cyberattacks or bioweapons (Barrett et al., 2023; Li et al., 2024), and reducing the production of offensive, biased, or misleading content (Lu et al., 2022; Yu et al., 2023; Yao et al., 2023). Current unlearning approaches include model optimization-based methods (Ilharco et al., 2022; Liu et al., 2022a; Yao et al., 2023; Eldan & Russinovich, 2023; Jia et al., 2024; Zhang et al., 2024a; Li et al., 2024) and input prompt or in-context learning-based techniques (Thaker et al., 2024; Pawelczyk et al., 2023; Liu et al., 2024a). However, many lack effectiveness, leading to either under-forgetting or over-forgetting, as shown by recent LLM unlearning benchmarks such as TOFU for fictitious unlearning (Maini et al., 2024) and MUSE for private or copyrighted information removal (Shi et al., 2024). Recent studies also show that even after unlearning, models can remain vulnerable to adversarial attacks (Schwarzschild et al., 2024; Patil et al., 2024; Lynch et al., 2024) or relearning from a small number of data (Hu et al., 2024; Lynch et al., 2024). This evidence suggests that effective unlearning for LLMs is far from trivial. Among current efforts, NPO (negative preference optimization) (Zhang et al., 2024a) stands out as a promising method, offering key advantages such as a bounded unlearning loss and gradient ascent with weight smoothing to enhance stability and control. However, we will show that the advantages of NPO can be limited by the presence of reference model bias (Sec. 4).

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

 Probability on $\mathcal{D}_f$ 
 Rouge-L on $\mathcal{D}_f$ 
 Truth ratio on $\mathcal{D}_f$ | $\uparrow$ 
 $\downarrow$ 
 $\downarrow$ 
 $\uparrow$ | Model utility 
 ( harmonic mean of 9 utility metrics) 
 Probability on $\mathcal{D}_r/\mathcal{D}_{\mathrm{real\_author}}/\mathcal{D}_{\mathrm{world\_facts}}$ 
 Rouge-L on $\mathcal{D}_r/\mathcal{D}_{\mathrm{real\_author}}/\mathcal{D}_{\mathrm{world\_facts}}$ 
 Truth ratio on $\mathcal{D}_r/\mathcal{D}_{\mathrm{real\_author}}/\mathcal{D}_{\mathrm{world\_facts}}$ | $\uparrow$ 
 $\uparrow$ 
 $\uparrow$ 
 $\uparrow$ |
| MUSE | LLaMA-2 7B 
 ICLM-7B | Unlearning real-world knowledge from texts about Harry Potter and BBC News | KnowMem on $\mathcal{D}_f$ 
 VerbMem on $\mathcal{D}_f$ 
 PrivLeak | $\downarrow$ 
 $\downarrow$ 
 $\to 0$ | KnowMem on $\mathcal{D}_r$ | $\uparrow$ |
| WMDP | Zephyr-7B-beta | Unlearning hazardous knowledge from bio/cybersecurity texts | Accuracy on WMDP-Bio 
 Accuracy on WMDP-Cyber | $\downarrow$ 
 $\downarrow$ | Accuracy on MMLU | $\uparrow$ |

## 4  UNCOVERING REFERENCE MODEL BIAS: A LIMITATION OF NPO

In this section, we highlight a key weakness of NPO, which we term '*reference model bias*', and provide a concise description below. That is, the incorporation of the reference model in NPO biases the unlearning objective towards enlarging the distance relative to the reference model. This is analogous to inductive bias, where a machine learning algorithm is guided to favor certain patterns over others based on the underlying assumptions of the parametric model. Specifically, as noted in (2), minimizing the NPO loss drives $\pi_{\boldsymbol{\theta}}(y|x) \ll \pi_{\mathrm{ref}}(y|x)$. However, using $\pi_{\mathrm{ref}}$ as the basis for NPO's unlearning criterion can introduce negative effects, as illustrated by the limitation (L1)-(L2).

**(L1) NPO suffers from uneven allocation of unlearning power.** At first glance, driving $\pi_{\boldsymbol{\theta}}(y|x) \ll \pi_{\mathrm{ref}}(y|x)$ in NPO appears desirable for unlearning on the forget set, where the reference model $\pi_{\mathrm{ref}}$ is given by the initial model prior to unlearning. The potential issue is that NPO's reliance on $\pi_{\mathrm{ref}}$ can lead to an uneven allocation of unlearning power, even misaligned with the true sample-specific unlearning difficulty. We elaborate on this issue through two examples.

*(Example 1: Unlearning short vs. long-response data.)* In this example, we evaluate unlearning performance across different types of forget data points, categorized by their response lengths (*i.e.*, short vs. long). As noted in (Meng et al., 2024), a reference model may exhibit a bias toward generating *longer but lower-quality* sequences. Consequently, these low-quality long texts tend to be easier to unlearn compared to short-length forget data. This suggests that allocating additional optimization power to further enlarge the distance from the reference model for these easy-to-unlearn samples is unnecessary. Such an allocation leads to an uneven distribution of optimization power,

disadvantaging the unlearning of shorter-response forget data points (*i.e.*, harder examples). Indeed, **Fig. 1-(b)** shows that NPO exhibits a greater distance from Retrain when unlearning the top 50% shortest-length forget data, resulting in a lower FQ of $0.58$. In contrast, NPO performs better unlearning for the longer 50% of the forget set, yielding a higher FQ of $0.81$. Therefore, NPO stays ineffective at unlearning forget data with short responses. This issue is also illustrated in Fig. 1-(a), where over-forgetting easy examples in NPO can lead to under-forgetting hard examples. Further, it will be demonstrated using a mixture of Markov chains in Sec. 5.

*(Example 2: Unlearning strongly vs. weakly-memorized forget data.)* We next explain (L1) from the perspective of unlearning vs. data memorization. Consider two forget sets, $\mathcal{D}_{f,1}$ and $\mathcal{D}_{f,2}$, where $\mathcal{D}_{f,1}$ is more strongly memorized by the model than $\mathcal{D}_{f,2}$. As a result, the prediction loss on $\mathcal{D}_{f,1}$ is smaller, leading to a higher prediction probability $\pi_{\text{ref}}$. Accordingly, the NPO gradient smoothing term in (3) becomes smaller for $\mathcal{D}_{f,1}$, meaning NPO allocates less first-order optimization power to it. However, $\mathcal{D}_{f,1}$, being strongly memorized, should ideally receive more unlearning power. See **Table A1 of Appendix C.3** for experimental justification on the above example.

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

-response forget data may have lower quality (Meng et al., 2024), is closer to the unlearning boundary, and requires less intervention (Fig. 1-(a)). In the extreme case where $\beta \to 0$, the SimNPO's gradient reduces to a *weighted GA*: $\nabla_{\boldsymbol{\theta}} \ell_{\mathrm{SimNPO}}(\boldsymbol{\theta}) \to \mathbb{E}_{(x,y)\in\mathcal{D}_{\mathrm{f}}}[1/|y|\nabla_{\boldsymbol{\theta}} \log \pi_{\boldsymbol{\theta}}(y|x)]$. This is different from NPO, which becomes GA as $\beta \to 0$. **Fig. 3** empirically demonstrates the advantage of length normalization in SimNPO for unlearning. As shown, SimNPO outperforms NPO in both forget quality and model utility, coming closest to Retrain. Even in the special case where $\beta = 0$ (*i.e.*, Weighted-GradDiff), the length normalization provides benefits over the vanilla GradDiff baseline.

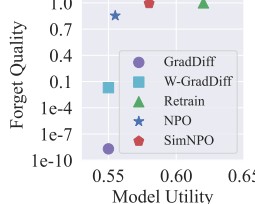

Figure 3: Forget quality vs. model utility on TOFU with forget set size of 5%. Weighted-GradDiff (W-GradDiff) is the variant of SimNPO at $\beta = 0$.

(b) In addition, the reference-free, length-normalized weight smoothing prevents early-stage ineffectiveness during unlearning. It can be shown from (5) that $w'_{\boldsymbol{\theta}}(x,y) < 2/|y|$, with the distribution of weights $w'_{\boldsymbol{\theta}}(x,y)$ depending on the specific forget data samples. This contrasts with NPO, where the weight distribution concentrated around $w_{\boldsymbol{\theta}}(x,y) \approx 1$ during the early unlearning stage. Extended from Fig. 2-(a)&(b), **Fig. 4** provides a detailed comparison between the gradient weights of SimNPO and NPO. As shown, SimNPO exhibits a much stronger correlation with the response length $|y|$ during the first two unlearning epochs, prioritizing short-length forget data that are initially harder to forget. At later epochs, the gradient weights become more uniform, reflecting that SimNPO can then treat different forget data with even optimization power. This trend is different from NPO, which assigns more uniform gradient weights early on and only accounts for data-specific difficulty when $w_{\boldsymbol{\theta}}(x,y)$ decreases in the later stages of unlearning. Therefore, similar to NPO, SimNPO benefits from gradient weight smoothing, enhancing unlearning stability. See **Appendix C.5** for the empirical

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

. 4. Furthermore, we remark that the advantages of SimNPO are not solely due to its awareness of forget data response length, as the MUSE dataset features forget data with equal response lengths. We further evaluate SimNPO's performance in hazardous knowledge unlearning on WMDP, as detailed in **Appendix C.8**.

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

We use TOFU Forget05 as the forget set $\mathcal{D}_f$, splitting it evenly into $\mathcal{D}_{f,1}$ and $\mathcal{D}_{f,2}$. The divided subsets $\mathcal{D}_{f,1}$ and $\mathcal{D}_{f,2}$ follow the same distribution of fictitious author information. We fine-tune the LLaMA-2 7B chat model on the original retain set of TOFU together with $\mathcal{D}_{f,1}$, *i.e.*, $\mathcal{D}_{retain} \cup \mathcal{D}_{f,1}$, to obtain the reference model before unlearning. The resulting reference model strongly memorizes $\mathcal{D}_{f,1}$ but least memorizes $\mathcal{D}_{f,2}$, despite both being drawn from the same distribution. We then perform unlearning using SimNPO and NPO over $\mathcal{D}_{f,1} \cup \mathcal{D}_{f,2}$. The unlearning performance, measured in terms of forget quality (FQ) and model utility, is presented in Table A1.

As shown in Table A1, since the original model was trained on $\mathcal{D}_{f,1}$, its prediction loss $-\log(\pi_{ref})$ on $\mathcal{D}_{f,1}$ is relatively small, leading to a higher prediction probability $\pi_{ref}$ on $\mathcal{D}_{f,1}$. Consequently, the NPO gradient smoothing term in (3) becomes relatively smaller for $\mathcal{D}_{f,1}$ due to the reference model's bias $\pi_{ref}$ on $\mathcal{D}_{f,1}$. As a result, NPO allocates less first-order optimization power to $\mathcal{D}_{f,1}$ (due to the smaller weight before the gradient) and focuses more on $\mathcal{D}_{f,2}$. This imbalance leads to better FQ

Table A1: Unlearning performance on differently memorized forget sets $\mathcal{D}_{f,1}$ and $\mathcal{D}_{f,2}$ in TOFU.

|  | FQ on $\mathcal{D}_{f,1}$ | FQ on $\mathcal{D}_{f,2}$ | Utility |
|---|---|---|---|
| Original | 0.00 | 0.00 | 0.62 |
| NPO | 0.32 | 0.69 | 0.56 |
| SimNPO | 0.70 | 0.72 | 0.59 |

for NPO on $\mathcal{D}_{f,2}$ compared to $\mathcal{D}_{f,1}$. However, $\mathcal{D}_{f,1}$ should ideally receive more unlearning power, as it was strongly memorized before unlearning. In contrast, SimNPO, by leveraging a reference-model-free reward, achieves a much smaller FQ difference between $\mathcal{D}_{f,1}$ and $\mathcal{D}_{f,2}$ while delivering higher FQ for both datasets compared to NPO. Furthermore, SimNPO demonstrates better model utility relative to NPO.

## C.4 Ablation Studies on SimNPO's Hyperparameter Selection

As shown in (4), $\beta$ and $\gamma$ are the two hyperparameters that control the unlearning effectiveness and utility preservation of SimNPO. Similar to NPO, $\beta$ is a temperature hyperparameter used to regulate the intensity of unlearning but normalized by the response length $|y|$ in SimNPO. As $\beta \to 0$, SimNPO approaches weighted GA in Fig. 3. $\gamma$ is the reward margin parameter from SimPO, which introduces a constant shift to the (per-sample) prediction loss $-(\beta/|y|)\log\pi_{\theta}(y|x)$ in SimNPO. Consequently, a larger $\gamma$ imposes a stricter unlearning margin, which could further suppress the model utility.

**Fig. A1-(a)** and **Fig. A1-(b)** illustrate the forget quality and model utility of SimNPO under various values of $\beta$ and $\gamma$ on TOFU forget05. The results show that when $\beta$ is too small or $\gamma$ is too large, forget quality tends to decrease towards zero. Additionally, for a fixed $\beta$, increasing $\gamma$ leads to lower model utility. Notably, setting $\gamma = 0$ consistently yields the best balance between unlearning performance and utility preservation across different $\beta$ values, which supports our choice of $\gamma = 0$ in SimNPO.

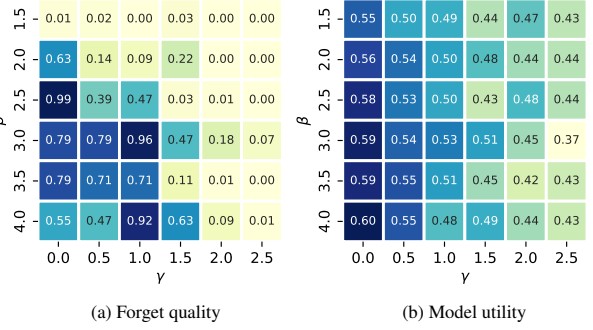

(a) Forget quality      (b) Model utility

Figure A1: Forget quality (a) and model utility (b) of SimNPO under different combinations of $\beta$ and $\gamma$ on TOFU Forget05.

## C.5 Further Explanation of Divergence

The term "divergence" refers to the optimization divergence from the pre-trained state, describing the process of deviating from the converged pre-trained model state to reverse the existing learning of the forgotten data. Thus, we measures the KL divergence on TOFU Forget05 between the unlearned model and the original model. The results, presented in **Fig. A2**, demonstrate that SimNPO amd NPO, exhibits that SimNPO achieves a logarithmic divergence rate against the unlearning steps $T$ like NPO in (Zhang et al., 2024a), as opposed to the linear divergence rate observed with GA.

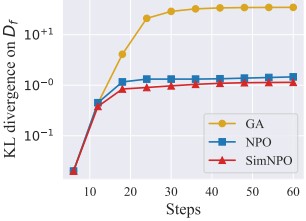

Figure A2: KL divergence between the unlearned and original model for GA, NPO and SimNPO on TOFU Forget05

## C.6 COMPUTATION COST

**SimNPO is more efficient than NPO.** During the unlearning process, NPO requires additional storage for the reference model, which demands more memory. Moreover, NPO needs to compute $\log(\pi_{\text{ref}}(y|x))$ at each step, resulting in higher time consumption. In contrast, SimNPO employs reference-free optimization, requiring less memory and time as shown in **Table A2**.

Table A2: Comparison of GPU memory and running time for Retrain, NPO and SimNPO on TOFU with forget size 5%.

| Method | Memory (GB) | Time (min) |
|---|---|---|
| Retrain | 20 | 120 |
| NPO | 27 | 36 |
| SimNPO | 21 | 25 |

## C.7 EXPERIMENTAL RESULTS ON MUSE

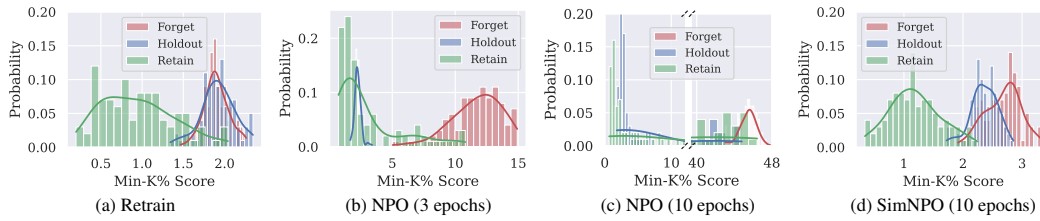

(a) Retrain    (b) NPO (3 epochs)    (c) NPO (10 epochs)    (d) SimNPO (10 epochs)

Figure A3: Distribution of Min-K% probability scores, a memorization metric used in MUSE applied to $\mathcal{D}_{\text{f}}$, $\mathcal{D}_{\text{r}}$, and a holdout set, respectively. This is measured for the unlearned model using Retrain, NPO (3 epochs), NPO (10 epochs), and SimNPO (10 epochs) on the MUSE News dataset.

We test the performance of SimNPO on MUSE News and Books datasets, with the results presented in **Table A3**.