# OpenReview forum: "Simplicity Prevails: Rethinking Negative Preference Optimization for LLM Unlearning"
_ICLR.cc/2025/Conference — Submitted to ICLR 2025_

### Official Review · Reviewer_Fdki · 2024-11-04

**Soundness:** 3
**Presentation:** 3
**Contribution:** 2
**Rating:** 6
**Confidence:** 4

**Summary:**

The paper proposes an innovative approach to Large Language Model (LLM) unlearning, called SimNPO, designed to remove undesirable data influences efficiently. This method advances the state-of-the-art Negative Preference Optimization (NPO) framework by addressing a key limitation: reference model bias. The authors introduce a reference-free approach, simplifying preference optimization without relying on a baseline model, which enhances unlearning robustness while retaining model utility. Extensive experiments on benchmarks like TOFU and MUSE demonstrate SimNPO's effectiveness in surpassing existing unlearning baselines, and it shows promise in defending against relearning attacks.

**Strengths:**

1. SimNPO provides a unique, reference-free framework that improves upon the limitations of NPO, specifically addressing reference model bias, making it a significant advancement in the field of unlearning for LLMs.

2. The approach is computationally efficient, allowing for effective unlearning without requiring full model retraining, which is resource-intensive and often impractical.

3. The method is tested on diverse datasets (TOFU, MUSE, WMDP), and results are detailed, showcasing the framework’s strong performance in preserving model utility and achieving high forget quality.

**Weaknesses:**

1. The technical contributions are sound but limited. The authors propose a reference-free optimization framework. However, optimization with a reference-free objective is not novel and has already been proposed by SimPO. Moreover, the length-normalized reward and reward margin parameter in SimNPO are also derived from SimPO.

2. SimNPO introduces two additional hyperparameters that must be manually adjusted to achieve optimal performance, significantly increasing the complexity and time cost of experiments. This greatly limits the potential application of this method for large language models in real-world settings.

3. Is there any theoretical analysis or intuition? What is the divergence rate of SimNPO compared to NPO? The current version of this paper only provides an experimental report without solid analysis.

4. A potential issue is over-reliance on the reference model; would adjusting the weight of the reference model help mitigate this problem? While removing the reference model entirely might solve the over-reliance issue, could it lead to insufficient performance for certain cases, such as math and coding datasets?

**Questions:**

Please see above

---

> ### Author Response · Authors · 2024-11-20
> **Look forward to your post-rebuttal feedback!**
>
> We appreciate Reviewer Fdki’s detailed review,, as well as the positive comments on the contributions and the overall presentation. Additionally, we value the constructive feedback provided. Our responses follow below, with [W] indicating weakness and [Q] for questions raised.

---

> ### Author Response · Authors · 2024-11-20
> **1. Response to W1 regarding limited technical contributions.**
>
> Thank you for raising this question. While SimNPO might appear straightforward by integrating SimPO into NPO, this does not diminish the technical contributions of our work. The reviewer expressed a concern that “However, optimization with a reference-free objective is not novel and has already been proposed by SimPO. Moreover, the length-normalized reward and reward margin parameter in SimNPO are also derived from SimPO.” We respectfully wish to clarify that **understanding the limitations of NPO caused by reference model bias**, offering insights into why a reference-free objective and length normalization are necessary, and demonstrating **how SimNPO effectively addresses NPO’s limitations** are far from trivial. This goes well beyond the mere application of SimPO in our proposal. Please see our detailed response in the general response **[GR3](https://openreview.net/forum?id=Pd3jVGTacT&noteId=8mSlEZ4vLU)**.

---

> ### Author Response · Authors · 2024-11-20
> **2. Response to W2 regarding the introduction of two additional hyperparameters  in SimNPO.**
>
> Thank you for raising this question. We apologize if there is any misunderstanding.
>
> Compared to the conventional NPO method, our proposed SimNPO introduces only one new hyperparameter, $\gamma$, as defined in Eq. (4). As explained in Lines 297–302, **the setting $\gamma = 0$ is preferred for unlearning optimization**, and we justified this choice with results presented in Fig. A1 (as cited in Line 302). This is because a larger $\gamma$ can exacerbate the utility drop during unlearning. With this setting, **SimNPO has only one hyperparameter, $\beta$**, which is the same as in NPO. Furthermore, a detailed ablation study on both $\gamma$ and $\beta$ is provided in **Appendix C.3 (Ablation Studies on SimNPO’s Hyperparameter Selection)**, as referenced in Line 417 (Experiment Setups). These studies demonstrate the sensitivity of SimNPO to hyperparameter selection, further justifying our selections.

---

> ### Author Response · Authors · 2024-11-20
> **3. Response to W3 regarding divergence rate and theoretical analysis of SimNPO.**
>
> This is an insightful question that inspires us to consider what kind of theoretical guarantees SimNPO might hold. One potential theoretical result we could establish is the slower optimization divergence rate achieved by SimNPO. Similar to NPO (Theorem 2, Zhang et al., 2024a), we think that **SimNPO could achieve a logarithmic divergence against the unlearning optimization iterations $T$**, as opposed to the linear divergence rate observed with gradient ascent (GA). In a broader context of theoretical analyses, the field of LLM unlearning currently lacks well-established frameworks or tools that can be readily employed by researchers, including ourselves, to derive rigorous guarantees on the optimization influence in this domain. While we acknowledge that deeper theoretical analysis would further strengthen our findings and is always desirable, this remains an open and important question for future research. We will incorporate this discussion into the conclusion and limitations sections of our paper to reflect the current gaps and potential directions for advancing the theoretical analyses of SimNPO and LLM unlearning.
>
> Inspired by the reviewer’s comment, we also conducted **additional experiments** to validate the slower divergence rate of SimNPO compared to GA. Specifically, we measured the **KL divergence on the forget set between the unlearned model and the original model**. The results, presented in **[Fig R2](https://ibb.co/TgcqWSR) in [GR2](https://openreview.net/forum?id=Pd3jVGTacT&noteId=1dU2FIB9C9)**, are consistent with our hypothesis that SimNPO, like NPO, exhibits a much slower divergence rate compared to the linear divergence rate observed with GA.

---

> ### Author Response · Authors · 2024-11-20
> **4. Response to W4 regarding adjusting the weight of the reference model.**
>
> This is an insightful suggestion. Generally speaking, if we can adjust the weight of the reference model to account for data-specific unlearning difficulty, it could serve as a potential solution to mitigate reference model bias. However, this approach comes with two potential challenges 1-2.
>
> 1.  Adjusting the weight of the reference model $\pi_{\mathrm{ref}}(y|x)$ to incorporate data-specific difficulty weights might require introducing and optimizing additional variables tied to individual data samples $(x, y)$. This could result in higher computational overhead, hampering the unlearning efficiency. On the other hand, if we use a data-agnostic universal weight for the reference model, its effectiveness in addressing the reference model bias becomes questionable.
>
> 2.  Attributing unlearning data difficulty: Accurately attributing unlearning data difficulty is a separate and complex problem. While having a data-wise attribution score for unlearning difficulty would allow us to better adjust the weight of the reference model, developing such a data attribution method is itself a challenging research problem. Nevertheless, we see potential avenues for addressing this, such as leveraging the data memorization perspective to estimate unlearning difficulty.
>
> We appreciate the suggestion and will incorporate the above discussion into Conclusion.

---

> ### Author Response · Authors · 2024-11-20
> **5. Response to W4 regarding insufficient performance by removing the reference model in certain cases, e.g, math datasets**
>
> The reviewer suggested investigating the possible side effects of removing the reference model in SimNPO on datasets beyond the standard unlearning benchmarks. We find this to be an insightful suggestion. To further validate the effectiveness and robustness of our method, we conducted **additional experiments** on math datasets, as per your suggestion. Below are the detailed experimental settings:
> - Forget Dataset: GSM8K, Composed of question-answer pairs based on basic arithmetic operations
> - Models: LLaMA-2-chat 7B and LLaMA-3-8B-Instruct
> - Unlearning methods: NPO and SimNPO
> - Evaluation metrics: GSM8K accuracy (to measure unlearning efficacy) and MMLU accuracy (to measure utility preservation)
>
> The results are presented in **Table R3** and **Table R4**. As observed, SimNPO achieves a lower GSM8K accuracy than NPO under the same MMLU, further demonstrating the effectiveness of our method. From this example, we infer that the side effect of removing the reference model appears to be minimal due to the following reasons: (1) Unlearning inherently operates as a divergence-driven optimization process (deviating from the pretrained model), implying that forgetting does not necessarily require using a reference model as the baseline. (2) From a utility perspective, SimPO has already shown that removing the reference model does not significantly harm utility. Therefore, the similar benefit  might hold for SimNPO. Based on the above, we feel that it is possible to remove the reference model in SimNPO without compromising unlearning performance or utility retention. We appreciate the reviewer’s suggestion and believe these findings further validate the robustness of our method.
>
>
> Table R3. Performance of LLaMA-2 7B on GSM8K
> |        | GSM8K | MMLU |
> |:------:|:-----:|:----:|
> | Origin |  0.24 | 0.46 |
> |   NPO  |  0.19 | 0.45 |
> | SimNPO |  0.08 | 0.45 |
>
>
> Table R4. Performance of LLaMA-3 8B on GSM8K
> |        | GSM8K | MMLU |
> |:------:|:-----:|:----:|
> | Origin |  0.75 | 0.64 |
> | NPO    |  0.26 | 0.63 |
> | SimNPO |  0.18 | 0.63 |

---

> ### Author Response · Authors · 2024-11-23
> **Thank you and look forward to following up.**
>
> We sincerely appreciate you taking the time to review our paper and provide valuable feedback. Based on your comments, we have provided individual responses, supplemented by general responses (GRs) where applicable, to address your questions on, e.g., [contributions](https://openreview.net/forum?id=Pd3jVGTacT&noteId=ntqPWnd6gu), [hyperparameter study](https://openreview.net/forum?id=Pd3jVGTacT&noteId=RHW87Plmqw), [divergence rate](https://openreview.net/forum?id=Pd3jVGTacT&noteId=lStyM4hCNG), and [suggested experiments](https://openreview.net/forum?id=Pd3jVGTacT&noteId=u8ZFZpIDzA).
>
> With four days remaining in the rebuttal period, please don’t hesitate to share any additional comments or questions. We will address them promptly and to the best of our ability. We would also greatly appreciate it if you could acknowledge the efforts we have made if you find our responses helpful.

---

> ### Author Response · Authors · 2024-11-27
> **Revised paper submitted and look forward to reviewer’s feedback**
>
> Dear Reviewer Fdki,
>
> We are pleased to inform you that a **revised version of our paper has been uploaded as a supplementary document**, with the original version left unchanged for ease of comparison. We have put significant effort into addressing your comments by providing both general and individual responses, conducting additional experiments, and revising the manuscript accordingly. We hope that our revisions have adequately addressed your concerns. During the extended author-reviewer discussion window, we remain available to respond to any follow-up questions you may have.
>
> Thank you very much for your time and consideration.

---

> ### Author Response · Authors · 2024-11-30
> **Look forward to your feedback!**
>
> Dear Reviewer Fdki,
>
> There are only 2 days remaining before the end of the discussion period. Following up on our earlier reminders, we sincerely look forward to your feedback. We have diligently worked to address your questions and concerns through our responses and revisions, and we humbly believe that we have resolved most, if not all, of the points you raised. If you find our responses and revisions satisfactory, we would greatly appreciate your acknowledgment of our efforts. However, if you have any additional concerns, we are more than happy to address them before the discussion deadline.
>
> Thank you for your time and consideration.
>
> Authors

---

### Official Review · Reviewer_HSyZ · 2024-11-08

**Soundness:** 3
**Presentation:** 2
**Contribution:** 2
**Rating:** 5
**Confidence:** 3

**Summary:**

This paper focuses on LLM unlearning, involving removing unwanted data influences or specific model capabilities from pre-trained models without needing to retraining from scratch. The authors state that NPO, the current state-of-the-art, suffers from reference model bias, and propose a new optimisation framework named SimNPO. SimNPO removes the dependence on the defence model by leveraging a reference-free optimisation method named SimPO, further normalising the loss for the varying lengths of strings. The authors conducted extensive experiments over a series of well established benchmarks, including TOFU, MUSE, WMDP, where the results show the superior performance of SimNPO over NPO and many other advanced methods.

**Strengths:**

The paper identifies and thoroughly analyzes the drawbacks inherent in NPO, seemingly providing some new insights.

By leveraging SimPO, SimNPO offers a simple yet effective solution that removes the reliance on a reference model, leading to more controlled and efficient unlearning.

Experiments show that SimNPO consistently achieves higher forget quality across various benchmarks, especially in scenarios with varying data difficulties.

SimNPO better preserves the model's utility on the retain set compared to NPO, maintaining performance on essential tasks.

**Weaknesses:**

“Divergence” seems to be an important terminology that is frequently mentioned throughout the manuscript, while I cannot find its clear definition. Similar issues appear for the terminology of "reference model bias". Do the authors indicate that the \pi_ref is inaccurate and thus the estimation of the weight in Eq 3 is inaccurate? Kindly please provide more detailed descriptions.

In L1 of Sec 4, the authors state that the reliance on \pi_ref lead to an uneven distribution of unlearning power. However,  it seems that \pi_ref allocates different attention for different strings: when we consider two strings with similar values of \pi_\theta, then the string with large \pi_ref will have smaller attention. Therefore, \pi_\theta can characterise the unlearning difficulty. The following example is also very weird, the authors assume a forget sample that has already been unlearned in \pi_ref. Generally, if a data point has been unlearned (or does not appear in the original model), why we further need unlearning.

I have another concern in L1, does the short string is generally harder to be unlearned than a long string? The validation experiments should be conducted across a broad range of current unlearning methods and difference metrics (as some metrics may have bias towards short / long strings). Personally, I think it is an important experiments because if the reverse is true, NPO can actually address the difficulties in unlearning long strings.

In L2, I wonder in what condition we can have an optimal gradient weight smoothing scheme. Also, does the proposed method can fulfil this goal?

I think the paper may some what overclaim its contribution. It seems that the main claim of this paper is that current methods might not be effective in handling short strings unlearning, motivating the paper to propose a new method that explicitly pay more attentions to those long strings. The current organisation of the paper is not clear, especially considering the fact that the authors do not discuss what kinds of the weighting mechanism is of our ultimate interest.

The authors claim that NPO suffers from over-unlearning. However, comparing the results in Table 3 between NPO and SimNPO for MUSE News, it seems that SimNPO faces the problem of under-unlearning.

To address L1 and L2, a simpler weighting strategy is \frac{1}{|y|}\cdot \pi_\theta (y|x), where  \frac{1}{|y|} handles L1 and \pi_\theta (y|x) handles L2. Compared with such a simple baseline, what are the superiority of the proposed method.

**Questions:**

Kindly please refer to the Weaknesses

---

> ### Author Response · Authors · 2024-11-20
> **Look forward to your post-rebuttal feedback!**
>
> We are grateful for the constructive comments from the reviewer, and provide our detailed responses below, where [W] represents weaknesses and [Q] stands for questions.

---

> ### Author Response · Authors · 2024-11-20
> **1. Response to W1 regarding divergence.**
>
> Thank you very much for sharing your concerns regarding the clarity of some of our terminologies, such as “divergence.” While we initially believed that some terms were standard in the context of our work, we now recognize that they may lead to confusion without additional elaboration. To address this concern and prevent any misunderstanding in the paper, we have reiterated key points from the original submission and provided further explanations for better clarity in our general response **[GR2](https://openreview.net/forum?id=Pd3jVGTacT&noteId=1dU2FIB9C9)**. In brief, the term “divergence” refers to optimization divergence from the pre-trained state, which describes the process of **deviating from the converged pre-trained model** to reverse the existing learning of the forgotten data, thereby achieving unlearning.

---

> ### Author Response · Authors · 2024-11-20
> **2. Response to W1 regarding reference model bias.**
>
> We apologize if our original presentation lacked clarity. Our intention was indeed for a better explanation by introducing the high-level concept of reference model bias using Fig. 1-(a), delving into its detailed implications in Sec. 4, and explaining how SimNPO addresses these limitations in Sec. 5.
> Reference model bias refers to the potential optimization issues that arise when the reference model is used as a basis for **unlearning optimization and gradient weight smoothing**, particularly when NPO drives the unlearning to widen the gap between the prediction probability of the unlearned model and the reference model on the forget set. In Sec. 4, we dissected the reference model bias into two key aspects: (L1): Unlearning optimization power allocation bias. (L2): Gradient weight smoothing bias. For more detailed responses, please refer to the general response **[GR1 part1](https://openreview.net/forum?id=Pd3jVGTacT&noteId=M51Ihx3cou)** and **[GR1 part2](https://openreview.net/forum?id=Pd3jVGTacT&noteId=Z9cYtRqSgN)**.

---

> > ### Comment · Reviewer_HSyZ · 2024-11-21
> >
> > Sorry, I cannot fully understand why the paper is motivated by the so-called "reference model bias".  Overall, there is not a clear definition in your manuscript (kindly please correct me if I make mistakes) about what the "reference model bias" is exactly. From the terminology, I think it is quite easy to misunderstand this term as $\pi_{ref}$ making some inaccurate or biased predictions. However, from the discussion in Section 4, it seems that the authors want to claim that the inherent weighting mechanisms of NPO may make mistakes. Therefore, I think the adopted terminology can be quite confusing and misleading.
> >
> > From L1 in Section 4, the authors stated that "over-reliance on $\pi_{ref}$ may lead to an uneven distribution of unlearning power". However, I think we do not need to make an even distribution of unlearning power, as some data may make more contributions to the unlearning procedures than others. However, the authors further claim the "irrespective of the sample-specific unlearning difficulty", making me really confuse about which situation is preferred, allocating equal weights or not?
> >
> > Also presented in L1, the authors stated that if a forget sample has already been unlearned in $\pi_{ref}$, further pushing $\pi_{\theta}<\pi_{ref}$ in unnecessary. I think if a data point has been unlearned by the original model, then it should not even exist in the forget set. Also, if $\pi_{ref}$ is truly more small, then $\pi_{\theta}$ will also be small and the NPO will converge. Kindly please correct me if I made mistakes.
> >
> > For L2, for the first epoch, I agree with you that w=1. However, linear warmup is also typically used for the first epochs, and if you do some more experiments with early stopping, you may find that GA can actually work pretty well.

---

> ### Author Response · Authors · 2024-11-20
> **3. Response to W2 regarding the role of \pi_ref in NPO.**
>
> Thank you for your insightful question. Please see our responses below.
>
> First, the reviewer commented that “However, it seems that $\pi_\mathrm{ref}$ allocates different attention for different strings: when we consider two strings with similar values of $\pi_\theta$, then the string with large $\pi_\mathrm{ref}$ will have smaller attention. Therefore, $\pi_\theta$ can characterize the unlearning difficulty.”
>
> We agree with the reviewer that $\pi_\mathrm{ref}$ provides a kind of attention to different data points. However, in the context of NPO, the situation becomes more complex. The issue lies in the NPO objective, as clarified in Lines 220–223 and Lines 226–232. Specifically, the NPO loss Eq. (2) is bounded below by $0$, and minimizing it drives the unlearning to **widen the gap between $\pi_\theta$ and the reference model $\pi_\mathrm{ref}$ on the forget set** (i.e., $\pi_\theta \ll \pi_\mathrm{ref}$). This NPO’s optimization, which is based on the distance from the reference model, gives rise to the unlearning optimization power allocation issue. That is, **NPO focuses on enlarging this distance with the reference model indiscriminately (without considering the intrinsic unlearning difficulty of individual data points)**. This is why, in Fig. 1-(a), we aimed to illustrate that it is unnecessary to aggressively widen the gap with the reference model for “easy” examples (which are already near the unlearning boundary) compared to “hard” examples. Fig. 2 experimentally justifies this uneven unlearning power allocation by showing the disparity in performance for forget data with different response lengths. Additionally, Fig. 6 further validates this limitation of NPO and highlights the benefit of SimNPO.
>
> Second, the reviewer commented that “The following example is also very weird, the authors assume a forget sample that has already been unlearned in $\pi_\mathrm{ref}$. Generally, if a data point has been unlearned (or does not appear in the original model), why do we further need unlearning.” We believe there may be some misunderstanding about our work.
>
> Our example in Fig. 2 is not about re-unlearning data that has already been unlearned in $\pi_\mathrm{ref}$. Instead, it illustrates the reference model bias issue of NPO when unlearning across different types of forget data points, categorized by response length. The motivation is detailed in Lines 234–239, where the reference model (i.e., the original LLM before unlearning) may exhibit a bias toward generating longer but lower-quality sequences, as also noted in (Meng et al., 2024). These lower-quality texts could be easier to unlearn (have been closer to the unlearning boundary). And further exacerbating the gap between $\pi_\theta$ and $\pi_\mathrm{ref}$ for these easy samples causes the optimization power allocation issue, as illustrated in Fig. 1-(a). Namely, such an optimization power allocation results in suboptimal unlearning for hard, short-response forgotten data points. Indeed, Fig. 2 shows that NPO performs worse when unlearning the top 50% shortest-length forget data in TOFU, as evidenced by a greater distance from Retrain, compared to the longer 50% of the forget set. Fig. 6 (left) further validates this observation, reinforcing this issue.
>
> In Fig. 6 (right), we constructed a synthetic setting to demonstrate that SimNPO unlearns already-unlearned data less aggressively than NPO. (Licong: indeed, it is unnecessary for the model to unlearn data that have already been unlearned, and  aggressively unlearning the already-unlearned data might deteriorate the performance of unlearning not-yet-unlearned data.) This example was specifically designed to highlight the reference model bias issue in NPO under an extreme case. Additionally, we conducted a new experiment (**Table R1**) in the general response **[GR1 part2](https://openreview.net/forum?id=Pd3jVGTacT&noteId=Z9cYtRqSgN)**, showing that  the reference model bias issue also extends to strongly memorized data (rather than already unlearned data) from the reference model. Please kindly refer to GR1 for more details.

---

> > ### Comment · Reviewer_HSyZ · 2024-11-21
> >
> > I am concerned about your goal of unlearning. As I remember, [1] suggests two goals of unlearning, one for full removal and other for influence removal. I wonder which goal of unlearning is of your interest. If you prefer the goal of full removal, I think we need to ensure $\pi_{ref}$ approaching 0; if you prefer the goal of influence removal, I wonder how you can control that the resulting unlearned models can be quite resemble to that training without unlearning data.
> >
> > [1] Sijia et al. Rethinking Machine Unlearning for Large Language Models
> >
> >
> > From the authors' feedbacks, it seems that the unlearning procedure should stop when close to the decision boundary. I would like to raise two questions, what is the decision boundary for LLMs and can SimNPO fulfil this requirement?

---

> ### Author Response · Authors · 2024-11-20
> **4. Response to W3 regarding short and long samples.**
>
> Thank you for your insightful question. As we addressed in the previous response, the issue lies in the NPO objective, whose minimization requires to enlarge the gap between $\pi_\theta$ and the reference model $\pi_\mathrm{ref}$ on the forget set (i.e., $\pi_\theta \ll \pi_\mathrm{ref}$). This mechanism leads to uneven unlearning power allocation between short and long samples.
>
> **(Additional experiment for validation)** Our rationale is based on the observation that the reference model (i.e., the original LLM before unlearning) may exhibit a bias toward generating longer but lower-quality sequences, as noted in (Meng et al., 2024). These lower-quality texts are typically closer to the unlearning boundary, making them easier to unlearn for NPO. This observation can also be validated by analyzing the prediction loss of the reference model (see **[Fig. R2](https://ibb.co/sWdxW97)**): **the reference model's loss on short samples is smaller than that on long samples in TOFU Forget05**, indicating that shorter samples are better learned and, therefore, more difficult to unlearn. As a result, further enlarging the gap for these longer (easier-to-unlearn) samples relative to the reference model exacerbates the uneven optimization power allocation, disproportionately disadvantaged shorter examples. In this sense, **shorter examples become more challenging to unlearn using NPO**. Our experimental results, presented in Fig. 2 (TOFU unlearning task) and Fig. 6 (synthetic mixture of Markov chains setting), both  support this finding.
>
> **(Additional justification via gradient weight smoothing )** Inspired by the reviewer’s comment, we would like to provide additional justification from another perspective: the gradient weight smoothing mechanism in NPO, as defined in Eq. (3). Because the reference model's prediction confidence on short samples is higher than that on long samples (owing to the smaller loss for short samples), NPO assigns smaller gradient weights to short samples during the unlearning process (based on the smoothing formula in Eq. (3)). This imbalance in gradient weights results in lower forget quality on short samples compared to long samples, exacerbating the uneven unlearning performance on short vs. long samples .

---

> ### Author Response · Authors · 2024-11-20
> **5. Response to W4 regarding optimal weight smoothing.**
>
> Thank you for your question; it raises an important topic for discussion. We believe that an optimal gradient weight smoothing strategy should dynamically adjust weights based on **both the unlearning difficulty of individual samples and the progression of the unlearning process**. Specifically:
>
> - **S1**: For harder-to-unlearn examples, their weights should ideally be larger to ensure sufficient optimization power. Conversely, weights for easier-to-unlearn examples should be smaller to prevent unnecessary effort on already unlearned data during the unlearning optimization process.
> - **S2**: As the unlearning process progresses, the weights should decrease during the later steps to avoid over-divergence or model collapse.
>
> These principles ensure a desirable unlearning optimization process, avoiding both underperformance on harder examples and instability in the overall optimization. Our SimNPO's gradient weights achieve this goal to some extent, as shown in **Fig. 5**. During the initial stage of unlearning, such as at epoch 1, SimNPO assigns larger gradient weights to short samples (**satisfying S1**). As the epochs progress, all gradient weights gradually decrease (**satisfying S2**). However, the precise definition of an optimal weight smoothing strategy and the methods to achieve it remain open questions in the field, inviting future research efforts to address these challenges.

---

> > ### Comment · Reviewer_HSyZ · 2024-11-21
> >
> > From my understanding, NPO can fulfil S1 as it assign larger weights to those samples with small $\pi_{ref}$ and can fulfil S2 as w approaches 0 when $\pi_{ref}$ approaches 0. Therefore, I am concerning, if your suggestions are correct, NPO also has an optimal weighting mechanism.
> >
> > Also, on the question of which factors lead to the hardness of data, I think the authors could do experiments in separating data based on its length (as your SimNPO suggested) and the values of $\pi_{ref}$ to discern easy and hard samples. Then, the authors could do experiments to show that the length is a better indicator for the hardness.

---

> ### Author Response · Authors · 2024-11-20
> **6. Response to W5 regarding over-claimed contributions.**
>
> We respectfully disagree with the reviewer’s comment regarding an “over-claimed contribution” and the partial interpretation of our work: “It seems that the main claim of this paper is that current methods might not be effective in handling short strings unlearning, motivating the paper to propose a new method that explicitly pay more attentions to those long strings.”
>
> To clarify, the observation that current methods might not be effective in handling short strings unlearning is a representative piece of evidence for the limitation (L1) of using the reference model in Sec. 4. **However, this is not the entirety of our contributions**.
>
> First, the limitation (L1) of NPO can also be observed in unlearning tasks when the forget data have the same length, such as the MUSE benchmark. Besides that, we conducted an **additional experiment** to demonstrate NPO’s limitation (L1) from a data memorization perspective, rather than based on the forget data length. This analysis, included in **[GR1 part2](https://openreview.net/forum?id=Pd3jVGTacT&noteId=1dU2FIB9C9)** with the results in **Table R1**, shows that the reference model bias issue also extends to strongly memorized data in the reference model.
>
> Second, the gradient weight smoothing issue, elaborated in (L2), is another significant side effect of reference model bias. As described in Lines 260–264, because the initially unlearned model $\boldsymbol{\theta}$ is derived from the reference model, the gradient weight smoothing in NPO lacks effectiveness during the early unlearning epochs, where $w_{\boldsymbol{\theta}}(x, y) \approx 1$, as shown in Fig. 3-(a, b). This limitation causes NPO to behave similarly to the conventional gradient ascent (GA) method during early optimization, as also noted in our previous response.
>
> Last, we would like to point out that SimNPO provides a unifying solution to address both (L1) and (L2), which we believe is a crucial contribution. In Sec. 5, we provide **point-by-point** insights into how SimNPO resolves these limitations. Please see **[GR3](https://openreview.net/forum?id=Pd3jVGTacT&noteId=8mSlEZ4vLU)**  for a detailed discussion on our contribution.

---

> ### Author Response · Authors · 2024-11-20
> **7. Response to W6 regarding the under-forgetting on MUSE.**
>
> Thank you for your insightful question. As a matter of fact, SimNPO does not exhibit under-unlearning in the MUSE News dataset, as **its PrivLeak is 11.90, which is greater than 0**. This observation is further corroborated by SimNPO’s results in Fig. 7-(d) compared to Retrain in Fig. 7-(a). When comparing the Min-K% Score of SimNPO on the forget set, it also deviates from the holdout set, indicating that it does not exhibit under-unlearning. However, the gap between SimNPO and Retrain is much smaller compared to NPO, highlighting SimNPO’s improved performance. In fact, across both the MUSE News and Books datasets, SimNPO achieves a PrivLeak value closest to 0 among all unlearning methods tested, showing that SimNPO is the unlearning method closest to Retrain.

---

> ### Author Response · Authors · 2024-11-20
> **8. Response to W7 regarding a simpler weighting strategy $\frac{1}{|y|}\cdot \pi_\theta (y|x)$.**
>
> Thank you for your valuable feedback. The suggested weighting strategy appears to align with the **length-normalized weighted GA** scheme that we discussed in Sec. 5, Lines 322–327, referred to as Weighted-GradDiff in **Fig. 4**. While we acknowledge that this simple weighting scheme demonstrates an improvement in forget quality over the vanilla GradDiff (as shown in Fig. 4), there **still remains a substantial performance gap compared to NPO or SimNPO**. This discrepancy arises because the simple weighting strategy can be regarded as a special case of SimNPO when $\beta \to 0$. Yet, in this case, the benefit of using preference optimization-type unlearning objective over the forget set is lost (e.g., weight smoothing), resulting in worse performance than NPO and SimNPO.

---

> ### Author Response · Authors · 2024-11-22
> **Further Response on W1 regarding reference model bias (Part 1)**
>
> **Thank-you note before the formal response:** Thank you very much for your prompt response and for raising a series of insightful questions. We deeply appreciate your feedback! Please find our further clarifications and detailed responses below.
>
> **Formal response:** We regret to hear that the concept of reference model bias was not fully conveyed. Please find our further responses to the different sub-questions below.
>
> (1) First, as stated in Lines 218–224, the reference model bias arises from the minimization of NPO’s objective function, which drives the optimized unlearning to maximize the distance between the prediction probability and the reference model on the forget set, towards $\pi_{\boldsymbol{\theta}} \ll \pi_{\mathrm{ref}}$. The bias lies in using the reference model as a distance basis for the unlearning criterion in NPO, which further leads to the problems described in (L1) and (L2). This explanation in Lines 218–224 was intended to such a concept of reference model bias before delving into its details regarding (L1) and (L2) in Sec. 4. To enhance clarity, we will explicitly provide a more precise and concise definition of reference model bias at the beginning of Sec. 4 and ensure it is properly introduced in the Introduction as well.
>
> (2) Second, thank you for highlighting potential confusion in our terminology. We are happy to revise it as mentioned above. However, we would like to clarify our rationale for using the term “reference model bias.” As previously explained, the introduction of the reference model in NPO makes the unlearning objective favor enlarging the distance relative to the reference model. This is analogous to “inductive bias,” where an ML algorithm is guided to favor one pattern over another based on the underlying assumptions from the parametric model. This reasoning underpins our use of the term, as described in Lines 218–220: “The reference model $\pi_{\mathrm{ref}}$ is used in NPO to measure and control the divergence speed required for unlearning.”
>
> (3) Third, we agree that some data contribute more to the unlearning process than others. However, the way these contributions are handled in NPO for unlearning is problematic, resulting in uneven (or even incorrect) optimization power allocation across different types of forget samples. Below, we provide more detailed justifications:
> - (a) While it is acceptable for the optimizer to further enlarge the gap for easy examples relative to the reference model, NPO comes at the cost of reduced optimization power for harder examples, given the total optimization budget. That is, in NPO, over-forgetting easy examples can result in under-forgetting hard examples, as further supported by the evidence discussed in (b) below. Such an issue is illustrated in Fig. 1-(a) and referenced in Line 238. Our experimental validations further support this point, as demonstrated in Fig. 2 (or Fig. A2 in Appendix), which examines data difficulties based on response length, and in Fig. 6, which evaluates a synthetic setup. For additional details, please refer to our general response [GR1 part2](https://openreview.net/forum?id=Pd3jVGTacT&noteId=Z9cYtRqSgN).
> - (b) We can also justify the above problem (b) from the perspective of gradient weight smoothing vs. data memorization. Consider two forget sets, $D_{f1}$ and  $D_{f2}$, where $D_{f1}$​ is more strongly memorized by the model than $D_{f2}$. The prediction loss$-\log (\pi_{\text{ref}})$ on $D_{f1}$ ​ is relatively small, resulting in a higher prediction probability $\pi_{\text{ref}}$ on $D_{f1}$. Consequently, the NPO gradient smoothing term in Eq. (3) becomes relatively smaller for $D_{f1}$ due to the reference model prediction $\pi_{\text{ref}}$ on $D_{f1}$ is higher. As a result, NPO allocates less first-order optimization power to $D_{f1}$ (due to smaller weight before gradient) and focuses more on $D_{f2}$. However, $D_{f1}$ should ideally receive more unlearning power since it was strongly memorized before unlearning. To validate this, we conducted experiments using TOFU dataset, as described in [GR1 part2](https://openreview.net/forum?id=Pd3jVGTacT&noteId=Z9cYtRqSgN) (“Additional experiment to improve the motivation of reference model bias”), demonstrating the issue of reference model bias in handling data with different levels of memorization.
>
> The above justifies why NPO overlooks the true sample-specific unlearning difficulty during optimization. We will revise our statement to make this point clearer.

---

> > ### Comment · Reviewer_HSyZ · 2024-11-24
> >
> > I am sorry, I still have some concerns about the "inductive bias". As claimed by the authors, $\pi_\theta<\pi_{\theta_{ref}}$ lead to the inductive bias. However, $\pi_\theta<\pi_{\theta_{ref}}$ is exactly what we want for LLM unlearning, most of the methods will achieve this goal, which makes me feel confused. Also, since $\pi_\theta<\pi_{\theta_{ref}}$ is a general (and beneficial) scenario, it raises the question of why it will lead to the mentioned problems of L1 and L2.

---

> ### Author Response · Authors · 2024-11-22
> **Further Response on W1 regarding reference model bias (Part 2)**
>
> (4) Fourth, the reviewer commented, “I think if a data point has been unlearned by the original model, then it should not even exist in the forget set.” We believe there might be a misunderstanding. Unlearning is a dynamic optimization process, and we did not imply that the data had been unlearned prior to the optimization. Instead, we meant that if an easy sample has been effectively unlearned at a certain stage of optimization, further pushing $\pi_{\boldsymbol \theta} \ll \pi_{\mathrm{ref}}$ becomes unnecessary and inefficient. This is illustrated in Fig. 1-(a), where an easy sample, which was not unlearned by $\boldsymbol \theta_{\mathrm{ref}}$ before unlearning, becomes unlearned under NPO, but the hard sample still remains (un)unlearned under NPO.
>
> (5) Fifth, Regarding your points on linear warmup and GA, we would like to clarify the following:
> - We have used linear warmup during the unlearning process for all unlearning methods.
> - GA, due to its unbounded loss and the lack of weight smoothing, leads to the issue of catastrophic collapse. Regardless of whether early stopping is applied, its performance remains suboptimal; see Fig 2 and Sec. 3.1 of the NPO work (Zhang et al., 2024a).
> For GA, even when early stopping is applied before model utility completely drops (see epoch 2 in **[Fig.R3](https://ibb.co/cXgbZ3D)**), its forget quality stays small of value 4.73e-15, indicating ineffectiveness of unlearning. This is consistent with Fig 2 (Zhang et al., 2024a).
>
> We hope this clarifies the points raised in your review. Please let us know if further elaboration is needed.

---

> > ### Comment · Reviewer_HSyZ · 2024-11-24
> >
> > For W4, I have a little concern. If $\pi_\theta \ll\pi_{ref}$ and the $\pi_{ref}$ itself is large, we will have the overall value of the npo weight approaches 0. It means that NPO also has some "early stopping" mechanism that is quite similar to the proposed method. Also, it seems that this point cannot be used to motivate the attentions via weights on those long strings. Overall, I think the logics of this paper is somewhat confusing and I also see the confusion in the comments of our reviewers. Therefore, if your paper is not accepted this time, I sincerely hope the authors to further polish the motivation and the logics of this manuscript.

---

> ### Author Response · Authors · 2024-11-22
> **Further response to W2 regarding the role of \pi_ref in NPO**
>
> Thank you for your insightful questions.
>
> The goal of unlearning in our work aligns more closely with data influence removal. This can be seen from our adoption of the recognized gold standard for unlearning, “Retraining from scratch” (i.e., Retrain), which is also considered for TOFU and MUSE unlearning benchmarks. In this context, an effective unlearning method should produce outputs closely resemble those of a retrained model trained on the dataset excluding the forget set. To evaluate this, our work measures the gap between the unlearning method and Retrain across several dimensions. (1) Truth ratio distribution on TOFU: We assess the degree of overlap in the truth ratio distribution between the unlearned and retrained models, as shown in Fig. A2 of the Appendix. (2) Min-K% probability distribution on MUSE: We compare the Min-K% probability distributions of the unlearned model on the forget, retain, and holdout sets against Retrain, as presented in Fig. 7. (3) Generated example similarity: We also visualize the similarity of generated content between the unlearned and retrained models, as shown in Table A2 of the Appendix.
>
> This is a very insightful question regarding the precise characterization of the decision boundary. To the best of our knowledge, this remains an open problem and a worthwhile area for further exploration. In our work, we believe it is appropriate to assess whether the unlearning decision boundary is approached by using Retrain as the ground truth. As noted earlier, we employed various metrics to evaluate the gap between an approximate unlearning method and Retrain, and we use the closest distance to Retrain as a key metric to determine whether SimNPO more effectively fulfills the unlearning task compared to NPO. For instance, as shown in Fig. 7d vs. Fig. 7a, SimNPO demonstrates better alignment with Retrain when evaluated using the distribution of data memorization scores across the forget, retain, and holdout sets. This provides evidence that SimNPO better approximates the ground truth unlearning decision boundary (provided by Retrain) than NPO.

---

> > ### Comment · Reviewer_HSyZ · 2024-11-24
> >
> > I wonder if the weighting mechanism of your proposed method can ensure influence removal, maybe with some heuristical or theoretical justifications. Just in my opinion, the goal of influence removal is quite challenging, as we may easily face the dilemma  between under-unlearning and over-unlearning. BTW, I also think the exact definition for the "boundary" of generative models are unclear.

---

> > > ### Author Response · Authors · 2024-11-24
> > > **Further response to influence removal and boundary.**
> > >
> > > Thank you for your thoughtful question. We agree that achieving effective data influence removal is a challenging task. **However, current benchmarks like TOFU and MUSE use Retrain as the gold standard to evaluate data influence removal, particularly from a membership inference perspective.** This is because Retrain provides an exact unlearning approach by not revealing forget data membership [R1]. This is why TOFU defines forget quality as a statistical measure of the distance between the True Ratio distributions of a retrained model and an approximately unlearned model. Similarly, in MUSE, PrivLeak measures influence removal from a data privacy perspective by evaluating the distance between Min-K% probability distributions of Retrain and the approximately unlearned model.
> > >
> > > In our paper, we evaluated SimNPO’s performance using forget quality and truth ratio distributions in TOFU (Fig. 2, Fig. A2, Tab. 2) and PrivLeak and Min-K% probability distributions in MUSE (Tab. 3, Fig. 7), comparing SimNPO with NPO and Retrain. As demonstrated, SimNPO indeed achieves the closest performance to Retrain, indicating better influence removal through these empirical measures. This can also been seen in our earlier **[response](https://openreview.net/forum?id=Pd3jVGTacT&noteId=vWZFzx5L4Z)**.
> > >
> > > Additionally, we considered the reviewer’s question on potential tools for theoretically analyzing data influence removal. To the best of our knowledge, influence function could be a promising approach to characterizing data-wise removal influence, as it has been used in analyzing data influence in simpler discriminative models [R2; Proposition 1]. However, influence function has notable limitations: (1) It is computationally expensive due to the need for second-order information, and (2) it is imprecise due to inherent assumptions made for data influence evaluation. As such, developing theoretical guarantees for data influence removal in LLMs indeed remains a challenging and open problem in the field.
> > >
> > > > [R1] Thudi, Anvith, et al. "Unrolling sgd: Understanding factors influencing machine unlearning." 2022 IEEE 7th European Symposium on Security and Privacy (EuroS&P). IEEE, 2022.
> > > >
> > > > [R2] Jia, Jinghan, et al. "Model Sparsity Can Simplify Machine Unlearning." The Thirty-eighth Annual Conference on Neural Information Processing Systems. NeurIPS 2023, 2023.

---

> ### Author Response · Authors · 2024-11-22
> **Further response to W4 regarding optimal weight smoothing**
>
> Thank you for your insightful and meaningful questions.
>
> First, NPO does NOT fulfill S1. Consider hard-to-unlearn samples that exhibit high prediction confidence $\pi_{\text{ref}}$ before unlearning (e.g., strongly memorized data as shown in **Table R1** in [GR1 part2](https://openreview.net/forum?id=Pd3jVGTacT&noteId=Z9cYtRqSgN) or short-response data in **[Fig. R2](https://ibb.co/sWdxW97)**). According to the NPO loss in Eq. (3), the gradient weights could become smaller for samples with high $\pi_{\text{ref}}$. Yet, this allocation is problematic because hard samples are not prioritized during the unlearning optimization process compared to easier samples. This is counterintuitive, as harder samples inherently require more unlearning effort than easier ones. As explained in GR1 and in our [first response](https://openreview.net/forum?id=Pd3jVGTacT&noteId=0iMWXo1JYe) this round,  this issue stems from NPO when introduced using the reference model as the unlearning basis, where minimizing the NPO objective in Eq. (3) does not accurately reflect the true difficulty of the forget data. Thus, NPO fails to fulfill S1.
>
> Second, regarding the factors contributing to the hardness of data in unlearning, we believe there could be multiple confounding factors. For instance, response length is one factor we considered in the TOFU benchmark for forgetting long vs. short data. However, in the MUSE dataset, the response length of the forget data is consistent due to its design. Despite this, we still observe SimNPO outperforming NPO, suggesting that other factors might contribute to the hardness of data. One possible factor is the degree of data memorization. This is also supported by our newly added experiments (see **Table R1** in [GR1 part2](https://openreview.net/forum?id=Pd3jVGTacT&noteId=Z9cYtRqSgN)), where we observe that NPO exhibits a bias in unlearning performance between strongly and weakly memorized data. Specifically, NPO performs worse in unlearning strongly memorized data compared to weakly memorized data. We are also open to incorporating additional validation experiments to further investigate the factors contributing to data hardness, as you suggested. If we are able to complete these experiments before the rebuttal deadline, we will post the results. Thank you for this insightful suggestion!

---

> ### Author Response · Authors · 2024-11-24
> **Further Response on additional experiments about unlearning short and long samples separately.**
>
> Thank you for your thoughtful and insightful questions. In response to your feedback, we conducted additional experiments for further validation. Specifically, we used TOFU Forget05 as the forget set $D_f$, splitting it evenly by length into $D_{short}$ and $D_{long}$. The divided subsets $D_{short}$ and $D_{long}$ follow the same distribution of fictitious author information as in TOFU. The original model was then unlearned with $D_{short}$ and $D_{long}$ separately. The results are presented in Table R4. We evaluated the performance using forget quality (FQ) on $D_{short}$ and $D_{long}$, as well as the average model utility when unlearning $D_{short}$ and $D_{long}$ respectively (Avg. Utility).
>
> Table R5: Performance of $D_{short}$ and $D_{long}$ on TOFU Forget05 using the LLaMA-2 7B chat model. The content format follows Table 2.
>
> |          | FQ on $D_{short}$ | FQ on $D_{long}$ |  Avg. Utility  |
> |:--------:|:-----------------:|:----------------:|:--------------:|
> |    NPO   |       0.28        |       0.37       |      0.52      |
> |  SimNPO  |       0.58        |       0.59       |      0.54      |
>
> As shown in Table R4, the performance of NPO has a larger variation between $D_{short}$ and $D_{long}$, with the forget quality for $D_{short}$ being lower than that for $D_{long}$. This indicates that $D_{short}$ is more challenging to unlearn compared to $D_{long}$ in NPO. In contrast, SimNPO demonstrates much smaller variance between $D_{short}$ and $D_{long}$, with forget quality consistently higher than NPO for both cases.

---

> ### Author Response · Authors · 2024-11-24
> **Further response to large $\pi_{\mathrm{ref}}$ and logics of the paper.**
>
> We agree that “If $\pi_{\boldsymbol \theta} \ll \pi_{\mathrm{ref}}$ and $\pi_{\mathrm{ref}}$ itself is large, the overall value of the NPO weight approaches 0.” **However, this "early stopping" does NOT necessarily result in effective unlearning, as it may halt at an incomplete unlearning stage where $\pi_{\boldsymbol \theta}$ is not sufficiently small but already exhibits a large distance from $\pi_{\mathrm{ref}}$, which has led to a small NPO weight.** This issue is precisely illustrated in the example of NPO's ineffectiveness in forgetting strongly memorized data compared to weakly memorized data, as discussed in [W1-Part-(3)-(b)](https://openreview.net/forum?id=Pd3jVGTacT&noteId=0iMWXo1JYe) and [GR1 Part 2](https://openreview.net/forum?id=Pd3jVGTacT&noteId=Z9cYtRqSgN) ("Additional experiment to improve the motivation of reference model bias"). In this case, strongly memorized data generates a sufficiently large $\pi_{\mathrm{ref}}$, which leads to a small smoothing weight and, consequently, halt at lacking unlearning optimization power for such examples. This behavior misaligns with the true data-wise unlearning difficulty. Overall, while $\pi_{\boldsymbol \theta} \ll \pi_{\mathrm{ref}}$ may occur, it does not provide a sufficient condition for effective unlearning in NPO since NPO could fail to truly reflect the unlearning difficulty of data during the dynamic optimization process.
>
> Regarding the comment “Overall, I think the logics of this paper is somewhat confusing and I also see the confusion in the comments of our reviewers. Therefore, if your paper is not accepted this time, I sincerely hope the authors to further polish the motivation and the logics of this manuscript.”   **We regret to hear this feedback and respectfully disagree with the concluding remark for the following reasons:**
>
> 1. While we acknowledge and value the reviewers’ suggestions to improve the clarity of our presentation, and have incorporated these improvements during the rebuttal phase, we do not believe the logic of the paper itself is confusing. We made substantial efforts to lay out NPO’s limitations and SimNPO’s point-by-point advantages, supported by both organized insights and experimental justifications. Despite some areas lacking initial clarity, we have ensured that all necessary information to establish a smooth and coherent logical flow was presented. In fact, much of this information was already included in the original paper; See our cited line numbers in our initial responses.
>
> 2. We view the rebuttal phase as an opportunity to address any misunderstandings or clarify areas of confusion. To that end, we have followed the reviewers’ suggestions, provided detailed responses, and revised the manuscript to improve clarity. We hope the reviewer recognizes our efforts in addressing questions and appreciates the improvements made during the rebuttal phase, especially when our responses have  addressed or alleviated your initial questions.

---

> ### Author Response · Authors · 2024-11-24
> **Further response to “inductive bias“.**
>
> We are sorry to hear that the reviewer still has concerns about reference model bias. At first glance, the condition $\pi_\theta < \pi_\text{ref}$ seems desirable for unlearning and is indeed considered by most methods. **However, the issue lies in how NPO applies this condition (as specified by the NPO's objective) during its dynamic optimization process.** As mentioned in our [response (point 3)](https://openreview.net/forum?id=Pd3jVGTacT&noteId=0iMWXo1JYe), NPO may further enlarge the gap for easier examples relative to the reference model, but this comes at the expense of reduced optimization power for harder examples. **This highlights the issue in the context of NPO: while $\pi_\theta < \pi_\text{ref}$ might seem valid as a condition, its usage in NPO fails to allocate unlearning power wisely based on the true unlearning difficulty of forget samples.** In [GR1 Part 2](https://openreview.net/forum?id=Pd3jVGTacT&noteId=Z9cYtRqSgN), we also validated through experiments comparing the unlearning performance of NPO against data memorization. Strongly memorized data are not effectively unlearned in NPO. Thus, this inductive bias from the reference model in NPO leads to L1 and L2.

---

> ### Author Response · Authors · 2024-11-30
> **Look forward to your feedback!**
>
> Dear Reviewer HSyZ,
>
> There are only 2 days remaining before the end of the discussion period. Following up on our earlier responses, we sincerely look forward to your feedback. We have diligently worked to address your questions and concerns through our responses and revisions. We hope that we have resolved most, if not all, of the points you raised. If you find our responses and revisions  helpful, we would greatly appreciate your acknowledgment of our efforts. However, if you have any additional concerns, we are more than happy to address them before the discussion deadline.
>
> Thank you for your time and consideration.
>
> Authors

---

### Official Review · Reviewer_ojtB · 2024-11-08

**Soundness:** 2
**Presentation:** 2
**Contribution:** 2
**Rating:** 5
**Confidence:** 4

**Summary:**

This paper focuses on large language model unlearning (LLM unlearning). Specifically, this work revisits a conventional optimization method, i.e., negative preference optimization (NPO), and identifies the issue of reference model bias that may hinder the performance of unlearning. To this end, this work proposes a new approach, namely, SimNPO, that extends NPO with a reference-free optimization framework like SimPO. Various experiments about the problem illustration and the performance evaluation of the newly proposed SimNPO are conducted to demonstrate the rationality of the proposal.

**Strengths:**

1. This work identifies one important issue with the conventional LLM unlearning method, NPO, regarding its reference model bias, which provides some insights into the learning behaviors of NPO on the allocation of unlearning power as well as the ineffective gradient weight smoothing.
2. It is reasonable that the SimNPO is promising in mitigating the reference model bias by considering the idea from SimPO to modify the original learning objective of NPO and the presentation also provides the insight discussion via the gradient derivation of SimNPO, as well as the further analysis of a mixture of Markov chains.
3. The experiments are conducted on three LLM unlearning benchmarks, which demonstrate the proposed method's general effectiveness.

**Weaknesses:**

1. Some statements and claims are not very clear and sometimes confusing. It would be better if the authors could improve the explanation or discussion for some words or sentences. The specific questions can refer to the Questions part.
2. The current presentation is not very good as it is difficult to understand some critical parts of the problem illustration or justification with long-distance cross-reference. For example, there are no related descriptions before or near the position where Fig.1 existed for the first time, and the corresponding caption can not fully explain all the meanings of the figure. Generally, it is hard to understand the major motivation (related to reference model bias) in the introduction part.
3. It seems that SimNPO is proposed based on NPO and adopts the SimPO to mitigate the reference model bias. The current presentation provides limited unique problems under the LLM unlearning scenario. In other words, SimPO is a reference-free preference optimization method, it can also mitigate the reference model bias if it exists in the framework of DPO. However, it is unclear about the unique novelty and insights of adopting SimPO to improve NPO in both perspectives of research question and methodology design.
4. The identified issue of NPO is reference model bias, while the reference model bias is not clearly illustrated in Section 4. For example, although we can get that both the forget quality and model utility of SimNPO would be better than NPO, there are only statistics of NPO at Figure 3 (a)-(b). Lacking of comparison makes it hard to intuitively get the meaning of blind allocation and ineffective gradient weight smoothing.

**Questions:**

1. I do not very understand what is the meaning of the statement "a principled optimization framework remains lacking" in the abstract. Although SimNPO revisited the NPO and improved it, would SimNPO be the principled optimization framework?
2. Could the author provide some explanation or illustration of the word "divergence" in line 55? Since it is italicized, is there any special definition or meaning for this word? And what is the meaning of "divergence-driven optimization methods"?
3. Could the author further explain the Figure 1(a)? The current version of the caption and description is hard to understand, and I can hard to get "overreliance on the reference model" as indicated in lines 76-77. It would be better if the authors could also provide some real examples instead of illustrations.
4. Are there any critical differences between the parallel improvement of SimPO from DPO and that of SimNPO from NPO? If not, does it mean that all the preference optimization methods can be applied in LLM unlearning?
5. Would the reference model bias just be an issue of a specific large language model? Could the issue generally be identified in different large language models?
6. I wonder whether there are results about statistical significance for the performance evaluation part?

---

> ### Author Response · Authors · 2024-11-20
> **Look forward to your post-rebuttal feedback!**
>
> Our sincerest thanks go to Reviewer ojtB for the detailed review of our paper. We appreciate the constructive questions shared, and we address each of them  below, using [W] to denote identified weaknesses and [Q] for the questions raised.

---

> ### Author Response · Authors · 2024-11-20
> **1. Response to W1, Q1, and Q2 on unclear statements and claims, such as "a principled optimization framework remains lacking" in abstract and "divergence" in line 55**
>
> Thank you very much for sharing your concerns regarding unclear statements and claims, along with the detailed questions in Q1 and Q2. We acknowledge that some of the terminology used in our submission might have led to confusion or misunderstandings. To address this, we would like to reiterate key points from the original submission and provide further explanations for clarity. While we initially believed these terms were standard and easy to digest in our context, we now realize they may cause confusion without additional elaboration, and we appreciate the opportunity to provide this clarification.
>
> **(Explanation of divergence)** The term **“divergence”** refers to optimization divergence from the pre-trained state, which describes the process of deviating from the converged pre-trained model state to reverse the existing learning of the data to be forgotten, thereby achieving unlearning. For detailed clarifications, please refer to the general response **[GR2](https://openreview.net/forum?id=Pd3jVGTacT&noteId=1dU2FIB9C9)**.
>
> **(Explanation of principled optimization)** The term **“principled optimization”** refers to technically grounded **optimization principles that LLM unlearning should ideally follow**. Before NPO, various LLM unlearning optimization objectives were proposed in the literature, most of which are heuristics-based. For instance, gradient ascent (GA)-type approaches have been widely used. However, **GA is not optimal** because it drives unlearning in the reverse direction of learning without controlled divergence, as reflected in its unbounded loss. **NPO addresses GA's unbounded loss issue** by leveraging a DPO framework, potentially offering two optimization benefits: (1) a bounded loss from below and (2) adaptive weight smoothing, as outlined in Lines 166-170. Thus, NPO has the potential to serve as a “principled” optimization framework for LLM unlearning. However, our work reveals that NPO falls short due to its limitations, (L1) and (L2), rooted in reference model bias, as detailed in Sec. 4. Building on these insights, Sec. 5 introduces SimNPO, which attempts to address these limitations with a point-by-point analysis of how SimNPO mitigates the issues identified in NPO.
>
> Given these considerations, this is why we claimed in the abstract that a principled optimization framework for LLM unlearning remains lacking in the literature. We hope this explanation clarifies our statements and strengthens the understanding of our contributions.

---

> ### Author Response · Authors · 2024-11-20
> **2. Response to W2 and Q3 regarding Fig.1-(a)**
>
> Apologies for any confusion caused by Fig. 1-(a), which was primarily included to visually explain the reference model bias in NPO and how SimNPO improves upon it.
>
> **(Further clarifications on Fig. 1-(a))** Recall that NPO unlearns by widening the gap between the unlearned model and the reference model, relying on the output of the reference model $\pi_{\text{ref}}$. However, $\pi_{\text{ref}}$ varies across different samples (**the green line in Fig. 1-(a)**). For some samples, $\pi_{\text{ref}}$ is already near the learn/unlearn boundary, making these "easy examples" for unlearning. In such cases, it is unnecessary to aggressively enlarge the gap with the reference model for these samples. Yet, NPO may continue to increase the distance between $\pi_{\theta}$ and $\pi_{\text{ref}}$ (**the blue line in Fig. 1-(a)**). Consequently, easy examples end up far beyond the boundary, while harder examples fail to cross it, as depicted in Fig. 1-(a). This behavior reflects NPO’s limitation: blind allocation of unlearning power, as highlighted in Line 238. This also corresponds to the benefit of SimNPO (Line 383).
>
> **(Existing experimental validation with Fig. 2)** Supporting Fig. 1-(a), we also provided experimental justification in Fig. 2 by analyzing unlearning performance across different types of forget data points, categorized by response length. The motivation for this analysis is elaborated in Lines 234-239, emphasizing that the reference model may exhibit a bias toward generating longer but lower-quality sequences, as noted in (Meng et al., 2024). These lower-quality, long texts could be easier to unlearn, further exacerbating the (unnecessary) optimization bias by enlarging the gap for these easy samples relative to the reference model. As a result, this allocation of optimization power is not even for unlearning shorter-response forgotten data points. Indeed, Fig. 2 demonstrates that NPO performs worse, as indicated by a greater distance from Retrain, when unlearning the top 50% shortest-length forget data compared to the longer 50% of the forget set.
>
> **(Existing experimental validation with Fig. A2)** In contrast, SimNPO avoids reference model bias and applies varying levels of unlearning power taking into account  the sample characteristics. This approach results in more uniform performance, as shown in Fig. A2, with **SimNPO achieving an FQ of 0.91 on short texts, significantly higher than NPO’s 0.58**. Additionally, Fig. 6, which presents results from a synthetic setup  using a mixture of Markov chains, further validates how SimNPO addresses NPO’s limitations.
>
> **(Revision action)** Finally, we acknowledge that while we explained reference model bias (referred to as “overreliance on the reference model” in our contributions at Lines 76-77) in the technical sections (4 and 5), this explanation is not sufficiently clear in the Introduction. We agree with the reviewer’s suggestion and plan to move Fig. A2 or the newly-conducted motivating example (**Table R1**, through the data memorization perspective) in **[GR1 part2](https://openreview.net/forum?id=Pd3jVGTacT&noteId=M51Ihx3cou)** to the Introduction to better illustrate Fig. 1-(a) and more effectively motivate our points. Thank you for your valuable feedback.

---

> ### Author Response · Authors · 2024-11-20
> **3. Response to W3 and Q4 regarding SimNPO’s novelty against SimPO and NPO**
>
> We regret to hear that the unique novelties of SimNPO were not clearly conveyed to the reviewer. Below, we provide some clarifications to better illustrate the unique problems that unlearning optimization and SimNPO aim to address:
>
> **(Difference between unlearning optimization and preference optimization)** Unlearning optimization, even when using NPO or SimNPO, differs fundamentally from ordinary preference optimization (PO), such as DPO and SimPO. **PO typically requires paired data**, including both positive (preferred) and negative (dispreferred) responses, to guide optimization. In contrast, **unlearning optimization**, which focuses on the forget objective, involves **only the forget data points**–dispreferred completions for the unlearned model.  This absence of paired data shifts the focus in **unlearning optimization to “optimization divergence”**, i.e., deviating from the already-trained stationary point (pre-trained model). Managing this divergence is critical to avoid over-forgetting (or model collapse) while maintaining retained model utility. This sensitivity to the optimization approach underscores the lack of a truly principled method for unlearning. For instance, gradient ascent (GA) is a known method that often leads to model collapse in unlearning tasks. The divergence-driven nature of unlearning optimization sets it apart from PO and applies equally to SimNPO compared to SimPO.
>
> **(The unique problems that SimNPO addresses in unlearning optimization)** As highlighted in Lines 165-170, NPO has the potential to better regulate the optimization divergence speed necessary for unlearning due to its key features: bounded loss from below and adaptive weight smoothing. **However**, as our work reveals in Sec. 4, NPO falls short because of its reference model bias limitations: (L1) Blind allocation of unlearning power across different kinds of forget samples, e.g., in terms of response length, with varying unlearning difficulties (see Fig. 2). (L2) Ineffectiveness of adaptive weight smoothing during early optimization stages (see Fig. 3). SimNPO, introduced in Sec. 5, provides point-by-point fixes to these limitations (see Fig. 4 and 5). These are unique challenges solved by SimNPO in the context of LLM unlearning. E.g, SimNPO is easier to unlearn short responses than NPO and unlearns already unlearned data less aggressively than NPO; as shown in Fig. 6. In the general response **[GR2](https://openreview.net/forum?id=Pd3jVGTacT&noteId=1dU2FIB9C9)**, we also empirically show that SimNPO shares the benefit of NPO in regulating the optimization divergence speed (see **[Fig. R1](https://ibb.co/TgcqWSR)** in **[GR2](https://openreview.net/forum?id=Pd3jVGTacT&noteId=1dU2FIB9C9)**).
>
> In summary, while SimPO improves upon DPO by removing the reference model, this improvement in the context of PO **cannot** be directly or easily transferred to the unlearning method comparison between SimNPO and NPO. This is because the divergence-driven nature of unlearning optimization introduces unique challenges and optimization behaviors that distinguish it from preference optimization. Therefore, our detailed analyses of NPO’s limitations and SimNPO’s ability to address these limitations represent novel contributions to the field.

---

> ### Author Response · Authors · 2024-11-20
> **4. Response to W4 regarding the reference model bias illustration and Fig.3 (a)-(b).**
>
> Thank you for your valuable feedback. However, we believe there may exist some misunderstandings regarding our insights and experimental justifications on reference model bias in Sec. 4 and Fig. 3 (a)-(b).
>
> In Sec. 4, as mentioned in Lines 221–224, we **dissect the reference model bias into two aspects**: (L1) unlearning optimization power allocation bias and (L2) gradient weight smoothing bias. Our goal was to provide a clear and detailed motivation and discussion of these points, rooted in the inductive bias of the reference model. We regret that these points may not have been conveyed effectively and sincerely apologize for any confusion. Please refer to  **[GR1 part1](https://openreview.net/forum?id=Pd3jVGTacT&noteId=M51Ihx3cou)** and **[GR1 part2](https://openreview.net/forum?id=Pd3jVGTacT&noteId=Z9cYtRqSgN)** for more clarification about (L1) and (L2).
>
> Fig. 3 (a)-(b) were used to explain the reference model bias **from the perspective of the gradient weight smoothing in NPO**, i.e., $w_{\boldsymbol \theta}(x,y)$ in Eq. 3. As described in Lines 260–264, since the initially unlearned model $\boldsymbol \theta$ is given by the reference model, the gradient weight smoothing lacks effectiveness during early unlearning epochs, as $w_{\boldsymbol \theta}(x,y) \approx 1$, as shown in Fig. 3-(a, b). This limitation causes **NPO to behave similarly to the conventional gradient ascent (GA) method during early optimization**, potentially leading to over-forgetting, a well-documented issue for GA. This behavior is further illustrated in Fig. 3-(d), which shows a significant utility drop during the early optimization phase. We want to emphasize again that Fig. 3-(a, b) is intended to provide statistics highlighting the limitation (L2) of gradient weight smoothing at the early optimization stage. Together with (L1), these two aspects offer a comprehensive view of NPO’s reference model bias-induced limitations.
>
> We hope the above  clarification provides a clearer perspective on our analyses on the reference model bias. A side note: In our general response **[GR1 part2](https://openreview.net/forum?id=Pd3jVGTacT&noteId=Z9cYtRqSgN)**, we also provided an additional motivating experiment (**Table R1**) from the  data memorization perspective to motivate the reference model bias issue.

---

> ### Author Response · Authors · 2024-11-20
> **5. Response to Q5 regarding different models.**
>
> **(Additional experiments with larger model)** Following the reviewer’s suggestion , we conducted **additional experiments** with a larger model LLaMA-3 8B to further validate the reference model bias, when unlearning the GSM8K dataset. As shown in **Table R2**, under the similar model utility (MMLU), SimNPO achieves lower accuracy on GSM8K, indicating more effective unlearning.
>
> Table R2. Performance of LLaMA-3 8B on GSM8K
> |        | GSM8K | MMLU |
> |:------:|:-----:|:----:|
> | Origin |  0.75 | 0.64 |
> | NPO    |  0.26 | 0.63 |
> | SimNPO |  0.18 | 0.63 |
>
>
> **(Clarification on existing models)** On the other hand, we would like to clarify that our study has covered a diverse range of model types, as outlined in Table 1, corresponding to different unlearning benchmarks. These models include LLaMA-2-chat 7B, LLaMA-2 7B, ICLM-7B, Zephyr-7B-beta, and GPT-2 (referenced in Line 367, Page 7).

---

> ### Author Response · Authors · 2024-11-20
> **6. Response to Q6 regarding statistical significance.**
>
> Thank you for your valuable feedback. We commit to including the standard deviations of all our results in the revised version for greater clarity. We would also like to note that all our experiment results have been reported based on the averaging performance over five independent random trials (as mentioned in Line 434, Page 9).

---

> ### Author Response · Authors · 2024-11-23
> **Thank you and look forward to following up.**
>
> Thank you very much for reviewing our paper and providing insightful and valuable comments. We have provided individual responses, supplemented by general responses (GRs) where applicable, to address your questions on, e.g., [unclear statements](https://openreview.net/forum?id=Pd3jVGTacT&noteId=MDkg0ZMyzF), [SimNPO’s novelty](https://openreview.net/forum?id=Pd3jVGTacT&noteId=XAeoL8CtzT), and [additional experiments](https://openreview.net/forum?id=Pd3jVGTacT&noteId=FqLRl7AThl). We hope you find our detailed responses clear and convincing.
>
> With four days remaining in the rebuttal period, please don’t hesitate to share any additional comments or questions. We will address them promptly and to the best of our ability. We would also greatly appreciate it if you could acknowledge the efforts we have made if you find our responses helpful.

---

> ### Author Response · Authors · 2024-11-27
> **Revised paper submitted and look forward to reviewer’s feedback**
>
> Dear Reviewer ojtB,
>
> We are pleased to inform you that a **revised version of our paper has been uploaded as a supplementary document**, with the original version left unchanged for ease of comparison. We have put significant effort into addressing your comments by providing both general and individual responses, conducting additional experiments, and revising the manuscript accordingly. We hope that our revisions have adequately addressed your concerns. During the extended author-reviewer discussion window, we remain available to respond to any follow-up questions you may have.
>
> Thank you very much for your time and consideration.

---

> ### Author Response · Authors · 2024-11-30
> **Look forward to your feedback!**
>
> Dear Reviewer ojtB,
>
> There are only 2 days remaining before the end of the discussion period. Following up on our earlier reminders, we sincerely look forward to your feedback. We have diligently worked to address your questions and concerns through our responses and revisions, and we humbly believe that we have resolved most, if not all, of the points you raised. If you find our responses and revisions satisfactory, we would greatly appreciate your acknowledgment of our efforts. However, if you have any additional concerns, we are more than happy to address them before the discussion deadline.
>
> Thank you for your time and consideration.
>
> Authors

---

### Official Review · Reviewer_x96s · 2024-11-12

**Soundness:** 3
**Presentation:** 2
**Contribution:** 2
**Rating:** 5
**Confidence:** 4

**Summary:**

The authors revisited the current unlearning optimization framework, negative preference optimization (NPO), and identified its reference model bias issue, which compromises unlearning effectiveness, particularly for forget data of varying difficulty. To address this, the authors introduced SimNPO, a simple yet effective framework that eliminates reliance on a reference model by leveraging simple preference optimization. The authors provided deep insights into SimNPO’s advantages through both synthetic data analysis and evaluations on existing unlearning benchmarks such as TOFU, MUSE, WMDP, and relearning attacks.

**Strengths:**

1. The authors revisit the NPO framework and identify its potential weakness–overreliance on the reference model–in LLM unlearning.

2. Building on insights into NPO’s limitations, the authors propose an improved LLM unlearning approach,
SimNPO, which extends NPO using a reference-free optimization framework.

3. The authors conduct extensive experiments to showcase the improvements of SimNPO over NPO across various unlearning benchmarks.

**Weaknesses:**

1.	Unclear Motivation: The primary issue lies in the unclear motivation of the paper. While it claims that the main limitation of NPO is the inductive bias of the reference model, Section 4 does not clearly convey this point, making it difficult for readers to understand.

2.	Missing Ablation and Sensitivity Analysis: The paper lacks both an ablation study and parameter sensitivity experiments, which are necessary to evaluate the impact of different components and parameter choices on the model’s performance.

3.	Limited Technical Contribution: The proposed SimNPO framework appears to be a straightforward combination of SimPO and NPO, with no substantial novel insights. Additionally, it lacks a deeper theoretical analysis to support its approach.

**Questions:**

See Weaknesses.

---

> ### Author Response · Authors · 2024-11-20
> **3. Response to W3 regarding Limited Technical Contribution**
>
> Thank you for raising this question. However, we respectfully disagree with the assessment that our work offers limited technical contributions and lacks substantial novel insights. Our primary goals in this work are indeed to **identify the potential limitations of NPO** and provide explanations of the underlying reasons (Sec. 4), as well as to **propose the corresponding fixes through SimNPO** and explain their rationale (Sec. 5). Please see our detailed clarifications in the general response **[GR3](https://openreview.net/forum?id=Pd3jVGTacT&noteId=8mSlEZ4vLU)**.
>
> We also appreciate the reviewer’s encouragement to explore deeper theoretical analyses. This is an insightful question that inspires us to consider the theoretical guarantees SimNPO might hold. One potential theoretical result we could establish is the slower optimization divergence rate achieved by SimNPO. Similar to NPO (Theorem 2, Zhang et al., 2024a), we think that **SimNPO could achieve a logarithmic divergence against the unlearning optimization iterations $T$**, as opposed to the linear divergence rate observed with gradient ascent (GA). In a broader context of theoretical analyses, the field of LLM unlearning currently lacks well-established frameworks or tools that can be readily employed by researchers, including ourselves, to derive rigorous guarantees on the optimization influence in this domain. While we acknowledge that deeper theoretical analysis would further strengthen our findings and is always desirable, this remains an open and important question for future research. We will incorporate this discussion into the conclusion and limitations sections of our paper to reflect the current gaps and potential directions for advancing the theoretical analyses of SimNPO and LLM unlearning.
>
> Inspired by the reviewer’s comment, we conducted **additional experiments** to validate the slower divergence rate of SimNPO compared to GA. Specifically, we measured the **KL divergence on the forget set** between the unlearned model and the original model. The results, presented in **[Fig. R1](https://ibb.co/TgcqWSR)**, are consistent with our hypothesis that SimNPO, like NPO, exhibits a logarithmic divergence rate, in contrast to the linear divergence rate observed with GA. These findings further support the theoretical foundation of SimNPO's optimization behavior.

---

> ### Author Response · Authors · 2024-11-20
> **2. Response to W2 regarding Missing Ablation and Sensitivity Analysis**
>
> Thank you for your valuable feedback. To avoid any misunderstanding, we would like to respectfully bring to the reviewer's attention that **we have conducted an ablation study** in **Appendix C.3 (Ablation Studies on SimNPO’s Hyperparameter Selection)**, as referenced in Line 417 (Experiment Setups).
>
> To be more specific, compared to the conventional NPO method, our proposed SimNPO introduces only one new hyperparameter, $\gamma$, as defined in Eq. (4). As explained in Lines 297–302, the setting $\gamma = 0$ is preferred for unlearning optimization, and we justified this choice with results presented in Fig. A1 (as cited in Line 302). This is because a larger $\gamma$ can exacerbate the utility drop during unlearning. With this setting, SimNPO has only one hyperparameter, $\beta$, which is the same as in NPO. A detailed ablation study on both $\gamma$ and $\beta$ has been provided in Appendix C.3 (Ablation Studies on SimNPO’s Hyperparameter Selection).

---

> ### Author Response · Authors · 2024-11-20
> **1. Response to W1 regarding unclear motivation on the inductive bias of the reference model and the difficult understanding of Sec. 4**
>
> Thank you for raising this important question. In Section 4, we aimed to provide a clear and detailed motivation and discussion regarding the limitations of NPO arising from the inductive bias of the reference model, specifically (L1) unlearning optimization power allocation bias and (L2) gradient weight smoothing bias. We regret to hear that these points may not have been clearly conveyed to the reviewer and sincerely apologize for any confusion. We kindly refer the reviewer to the general response **[GR1 part 1](https://openreview.net/forum?id=Pd3jVGTacT&noteId=M51Ihx3cou)** and **[GR1 part 2](https://openreview.net/forum?id=Pd3jVGTacT&noteId=Z9cYtRqSgN)** for detailed explanations. For convenience, we repeat the key points below to facilitate readability.
>
> **(Reiterating our key points in Sec. 4)**  First, we sincerely appreciate the opportunity to clarify the points that have been conveyed in our paper and to ensure they are communicated as effectively as possible.
>
> (L1): There could be an **optimization bias** introduced by the reference model. For example, as stated in Lines 231-233: If a forget sample $(x, y)$ has been close to unlearning for  the reference model $\pi_{\mathrm{ref}}(y|x)$, further pushing the unlearned model $\pi_{\boldsymbol \theta}(y|x) \ll \pi_{\mathrm{ref}}(y|x)$ may be unnecessary and inefficient. The illustration in Fig. 1-(a), as cited in Line 238, also clarifies this point: If a forget data point is close to the unlearning optimization boundary (i.e., an easy sample in Fig. 1-(a)), we should allocate less optimization power when unlearning this sample, rather than blindly enlarging the gap with the reference model. In addition to the schematic illustration in Fig. 1-(a), Fig. 2 provides experimental justification for (L1) by analyzing unlearning performance across different types of forget data points, categorized based on their response lengths. The motivation is elaborated in Lines 234-239, highlighting that the reference model may exhibit a bias toward generating longer but lower-quality sequences, as noted in (Meng et al., 2024). Consequently, these low-quality, long texts tend to be easier to unlearn. Accordingly, further enlarging the gap for these easier-to-unlearn points relative to the reference model makes  the optimization power allocation uneven for unlearning short-response forgotten data points. Indeed, Fig. 2 shows that NPO exhibits a worse unlearning performance, in terms of a greater distance from Retrain, when unlearning the top 50% shortest-length forget data than the longer 50% of the forget set.
>
> (L2) The other limitation caused by the reference model bias stays at **the gradient weight smoothing** scheme in NPO, i.e., $w_{\boldsymbol \theta}(x,y)$ in Eq. 3. This scheme is a major advantage of NPO. However, as illustrated in Lines 260-264, since the initially unlearned model $\boldsymbol \theta$ is given by the reference model itself, the gradient weight smoothing lacks effectiveness during the early unlearning epochs due to $w_{\boldsymbol \theta}(x,y) \approx 1$, as shown in Fig. 3-(a,b). A consequence of this limitation is that NPO behaves similarly to the conventional gradient ascent (GA) method during the early stages of optimization, potentially causing over-forgetting--an issue well-known for GA. This behavior is also justified in Fig. 3-(d), which depicts a significant utility drop during the early optimization phase. We would also like to note that the gradient weight smoothing results of SimNPO were omitted in Fig. 3-(a, b), as they are presented later in Fig. 5, where we systematically introduce SimNPO and highlight its gradient weight smoothing advantage over NPO.
>
> With the reiterated points (L1) and (L2) in Section 4, we sincerely hope that the reviewer will take another look at our original storyline and gain a clearer understanding of the motivation behind Sec. 4 compared to the first-round review.
>
> **(Improved motivation with newly conducted experiments)** We also appreciate the reviewer’s criticism, which prompted us to reflect on ways to strengthen our motivation regarding the reference model bias. Inspired by this feedback, we conducted a **new motivating experiment** to support our finding through a data memorization perspective.  Please refer to additional experiments (**Table R1**) in **[GR1 part2](https://openreview.net/forum?id=Pd3jVGTacT&noteId=Z9cYtRqSgN)** for more details.

---

> ### Author Response · Authors · 2024-11-20
> **Look forward to your post-rebuttal feedback!**
>
> We would like to express our gratitude to Reviewer x96s for the constructive questions raised. Our responses are organized with [W] marking weakness and [Q] for specific questions.

---

> ### Author Response · Authors · 2024-11-23
> **Thank you and look forward to following up.**
>
> Thank you for taking the time to review our paper. We sincerely appreciate your valuable feedback. We have provided individual responses to your comments, supplemented by general responses referenced where applicable, to address your questions on [motivation](https://openreview.net/forum?id=Pd3jVGTacT&noteId=aouVi1kVTz), [the ablation study](https://openreview.net/forum?id=Pd3jVGTacT&noteId=0cbuT3Cu9T), and [technical contributions](https://openreview.net/forum?id=Pd3jVGTacT&noteId=zXCzQUEhLG).
>
> With four days remaining in the rebuttal period, please don’t hesitate to share any additional comments or questions. We will address them promptly and to the best of our ability. We would also greatly appreciate it if you could acknowledge the efforts we have made if you find our responses helpful.

---

> ### Author Response · Authors · 2024-11-27
> **Revised paper submitted and look forward to reviewer’s feedback**
>
> Dear Reviewer x96s,
>
> We are pleased to inform you that a **revised version of our paper has been uploaded as a supplementary document**, with the original version left unchanged for ease of comparison. We have put significant effort into addressing your comments by providing both general and individual responses, conducting additional experiments, and revising the manuscript accordingly. We hope that our revisions have adequately addressed your concerns. During the extended author-reviewer discussion window, we remain available to respond to any follow-up questions you may have.
>
> Thank you very much for your time and consideration.

---

> ### Author Response · Authors · 2024-11-30
> **Look forward to your feedback!**
>
> Dear Reviewer x96s,
>
> There are only 2 days remaining before the end of the discussion period. Following up on our earlier reminders, we sincerely look forward to your feedback. We have diligently worked to address your questions and concerns through our responses and revisions, and we humbly believe that we have resolved most, if not all, of the points you raised. If you find our responses and revisions satisfactory, we would greatly appreciate your acknowledgment of our efforts. However, if you have any additional concerns, we are more than happy to address them before the discussion deadline.
>
> Thank you for your time and consideration.
>
> Authors

---

### Author Response · Authors · 2024-11-20
**GR3: Clarification on technical contributions**

Thank you for raising this question  (@Reviewers [x96s](https://openreview.net/forum?id=Pd3jVGTacT&noteId=YyI169NJBs), [ojtB](https://openreview.net/forum?id=Pd3jVGTacT&noteId=Njnkfs0Phh), [Fdki](https://openreview.net/forum?id=Pd3jVGTacT&noteId=3fxZV2nsCk)). While SimNPO might appear straightforward by integrating SimPO into NPO, this does not diminish the technical contributions of our work. We respectfully wish to clarify that understanding the limitations of NPO caused by reference model bias, offering insights into why a reference-free objective and length normalization are necessary, and demonstrating how SimNPO effectively addresses NPO’s limitations are far from trivial. This goes well beyond the mere application of SimPO in our proposal. Please see our detailed response below.

**(Identification of NPO’s current limitations in Sec. 4)** First, the identification of NPO’s current limitations is novel and not incremental. Prior to our work, NPO was increasingly recognized as an advantageous optimization framework for LLM unlearning due to its two key improvements–bounded loss from below and adaptive weight smoothing–over classical gradient ascent-based unlearning approaches, as detailed in Lines 165-187. However, we reveal that the advantages of NPO can be restricted due to the presence of reference model bias, which we thoroughly analyzed in Sec. 4. Specifically, we highlight two critical limitations: (L1) Blind allocation of unlearning power against different kinds of forget data points. (L2) Ineffectiveness of adaptive weight smoothing during early optimization phases. These insights into the limitations of NPO, supported by our analyses and experimental results in Sec. 4, are novel contributions to the field.

**(Point-by-point improvement and rationale by SimNPO in Sec. 5)** Second, our focus has been on providing a deep understanding of why the improved effectiveness of SimNPO exists and demonstrating its practical performance. In Sec. 5, under the subsection “Insights into SimNPO,” we conduct a point-by-point analysis of how SimNPO addresses NPO’s limitations (L1) and (L2) in Sec. 4. Additionally, we provide further insights through analyses and experiments using a mixture of Markov chains, which substantiate our insights into the benefits of SimNPO. These insights are non-trivial and are supported by our methodological analysis (see Eq. (5) and related discussions) and experimental validations (Figs. 4-6).

Last but not the least, we have tried our best to validate the effectiveness of SimNPO across **various unlearning tasks**, as acknowledged by reviewers x96s, HSyZ, and Fdki.

---

### Author Response · Authors · 2024-11-20
**GR2: Clarification on the term “divergence”**

Thank you very much for sharing your concerns regarding the clarity of some of our terminologies, such as “divergence.” (@Reviewers [ojtB](https://openreview.net/forum?id=Pd3jVGTacT&noteId=Njnkfs0Phh), [HSyZ](https://openreview.net/forum?id=Pd3jVGTacT&noteId=7Oox38ISD0), [Fdki](https://openreview.net/forum?id=Pd3jVGTacT&noteId=3fxZV2nsCk)). To address this and prevent any misunderstanding in the paper, we would like to reiterate key points from the original submission and provide further explanations for better clarity. While we initially believed that some terms were standard in the context of our work, we now recognize that they may lead to confusion without additional elaboration. We sincerely appreciate the opportunity to provide this clarification.

**(Explanation of divergence)** The term “divergence” refers to optimization divergence from the pre-trained state, which describes the process of **deviating from the converged pre-trained model state to reverse the existing learning of the forgotten data**, thereby achieving unlearning. In Lines 220-223, we provide one explanation of this concept: the divergence drives the optimization process to widen the gap between the prediction probability and the reference model on the forget set. In our paper, we overlooked the detailed definition of this term initially, as it is commonly used in the literature, for example, the over-divergence issue (i.e., model collapse) of gradient ascent (GA) identified in the NPO paper. To avoid potential confusion, we will clarify and explicitly define this term in the revised version of the paper.

**(Discussion on theoretical analysis for divergence rate and additional experimental validation)** This is an insightful question that inspires us to consider what kind of theoretical guarantees SimNPO might hold. One potential theoretical result we could establish is the slower optimization divergence rate achieved by SimNPO. Similar to NPO (Theorem 2, Zhang et al., 2024a), we think that **SimNPO could achieve a logarithmic divergence against the unlearning optimization iterations $T$**, as opposed to the linear divergence rate observed with gradient ascent (GA). In a broader context of theoretical analyses, the field of LLM unlearning currently lacks well-established frameworks or tools that can be readily employed by researchers, including ourselves, to derive rigorous guarantees on the optimization influence in this domain. While we acknowledge that deeper theoretical analysis would further strengthen our findings and is always desirable, this remains an open and important question for future research. We will incorporate this discussion into the conclusion and limitations sections of our paper to reflect the current gaps and potential directions for advancing the theoretical analyses of SimNPO and LLM unlearning.

Inspired by the reviewers’ comment, we conducted additional experiments to validate the slower divergence rate of SimNPO compared to GA. Specifically, we measured the **KL divergence on the forget set (TOFU Forget05) between the unlearned model and the original model**. The results, presented in **[Fig. R1](https://ibb.co/TgcqWSR)** are consistent with our hypothesis that SimNPO, like NPO, exhibits a logarithmic divergence rate, in contrast to the linear divergence rate observed with GA. These findings further support the theoretical foundation of SimNPO's optimization behavior.

---

### Author Response · Authors · 2024-11-20
**GR1: Clarification on Reference Model Bias and Improvements (Part 2/2)**

**(L2: Gradient weight smoothing bias caused by the reference model)** Another limitation caused by the reference model bias lies in the gradient weight smoothing scheme in NPO, $w_{\boldsymbol \theta}(x,y)$ in Eq. 3. This limitation is experimentally justified in Fig. 3. As described in Lines 260–264, since the initially unlearned model $\boldsymbol \theta$ is given by the reference model, the gradient weight smoothing lacks effectiveness during early unlearning epochs, as **$w_{\boldsymbol \theta}(x,y) \approx 1$**, as shown in Fig. 3-(a, b). This limitation causes NPO to behave similarly to the conventional gradient ascent (GA) method during early optimization, potentially leading to over-forgetting (and even model collapse), a well-known issue for GA. This behavior is further illustrated in Fig. 3-(d), which shows a significant utility drop during the early optimization phase.

**(Additional experiment to improve the motivation of reference model bias)** We also conducted a new motivating experiment to further support our claims from a data memorization perspective. Our rationale is to enforce the reference model to memorize certain forget data points and then investigate how the unlearning optimization responds to these strongly-memorized forget points compared to least-memorized forget points. Specifically, we use TOFU Forget05 as the forget set $D_{f}$, splitting it evenly into $D_{f1}$ and $D_{f2}$. Here the divided $D_{f1}$ and $D_{f2}$ follow the same distribution of fictitious author information in TOFU. We finetune the LLaMA-2 7B chat model on the original retain set of TOFU together with $D_{f1}$, i.e., $D_{\text{retain}} \cup D_{f1}$, to obtain the original model (i.e., the reference model) before unlearning. The resulting reference model then strongly memorizes $D_{f1}$ but least memorizes $D_{f2}$ although they are drawn from the same distribution of fictitious authors. We then perform unlearning using SimNPO and NPO over $D_{f1} \cup D_{f2}$. We present unlearning performance, in terms of FQ (forget quality) and model utility, in **Table R1**.

**Table R1: Performance of TOFU using the LLaMA-2 7B chat model. The content format follows Table 2.**

|          | FQ on $D_{f1}$ | FQ on $D_{f2}$ |  Utility  |
|:--------:|:--------------:|:--------------:|:----:|
| Original |      0.00      |      0.00      | 0.62 |
|    NPO   |      0.32      |      0.69      | 0.56 |
|  SimNPO  |      0.70      |      0.72      | 0.59 |

As shown in Table R1, since the original model was trained on $D_{f1}$, its prediction loss $-\log (\pi_{\text{ref}})$ on $D_{f1}$ is relatively small, leading to a higher prediction probability $\pi_{\text{ref}}$ on $D_{f1}$. Consequently, the NPO gradient smoothing term in Eq. (3) becomes relatively smaller for $D_{f1}$ due to the reference model bias $\pi_{\text{ref}}$ on $D_{f1}$. As a result, NPO allocates less first-order optimization power to $D_{f1}$ (due to smaller weight before gradient) and focuses more on $D_{f2}$. This imbalance leads to better forget quality (FQ) for NPO on $D_{f2}$ compared to $D_{f1}$. However, $D_{f1}$ should ideally receive more unlearning power since it was strongly memorized before unlearning. In contrast, SimNPO, by leveraging a reference-model-free reward, achieves a much smaller FQ difference between $D_{f1}$ and $D_{f2}$ and delivers higher FQ for both datasets compared to NPO. Furthermore, SimNPO demonstrates superior model utility relative to NPO, highlighting the effectiveness of our approach.

---

### Author Response · Authors · 2024-11-20
**GR1: Clarification on Reference Model Bias and Improvements (Part 1/2)**

Thank you for raising this question (@Reviewers [x96s](https://openreview.net/forum?id=Pd3jVGTacT&noteId=YyI169NJBs), [ojtB](https://openreview.net/forum?id=Pd3jVGTacT&noteId=Njnkfs0Phh), [Fdki](https://openreview.net/forum?id=Pd3jVGTacT&noteId=3fxZV2nsCk)). We apologize if our original presentation lacked clarity. We would like to kindly point out that our intention was to introduce **the reference model bias** issue at a high level in Fig. 1-(a) and then elaborate on it from **two key aspects** in Sec. 4: (L1) The issue of **unlearning power allocation**, where referencing the reference model fails to account for data-wise unlearning difficulty. (L2) The potential ineffectiveness of **gradient weight smoothing** when referenced to the reference model, particularly during early optimization stages. Given these analyzed limitations and their experimental justifications, we then elaborated on the one-to-one correspondence between the advantages introduced by SimNPO and the specific limitations it addresses. These improvements were further validated using a mixture of Markov chain analysis.

In the rebuttal, as reviewers expressed concerns regarding the clarity of the reference model bias, we provided further detailed explanations below to enhance the understanding of our work.


**(L1: Misallocation of unlearning power in NPO)** Recall that the minimization of NPO’s objective (Eq. (2)) drives the unlearning process by widening the gap between the unlearned model and the reference model, as explained in Lines 220–227. This process inherently relies on the sensitivity of the reference model $\pi_{\text{ref}}$ to the forget data samples. However, different samples elicit different responses from the reference model $\pi_{\text{ref}}$. For samples with a larger prediction loss, $-\log(\pi_{\text{ref}})$, the prediction probability $\pi_{\text{ref}}$ is smaller and often closer to the learn/unlearn boundary. Such samples are easier to unlearn and require less unlearning power compared to more challenging samples. However, in NPO, **even these easy samples may be assigned excessive power to unnecessarily enlarge the distance to the reference model**. This misallocation of unlearning power becomes uneven, disadvantaging the more difficult samples.

The presented Fig. 1-(a) and the discussion on (L1) in Sec. 4 were specifically designed to illustrate and reveal the above point. For instance, the reference model prediction $\pi_{\text{ref}}$ could vary across different samples **(as shown by the green line in Fig. 1-(a))**. For some samples, $\pi_{\text{ref}}$ is already near the learn/unlearn boundary, making these "easy examples" for unlearning in Fig. 1-(a). In such cases, aggressively enlarging the gap with the reference model for these samples becomes unnecessary. Yet, NPO may continue to increase the distance between $\pi_{\theta}$ and $\pi_{\text{ref}}$ **(as shown by the blue line in Fig. 1-(a))**. As a result, easy examples end up far beyond the boundary, while harder examples fail to cross it, as depicted in Fig. 1-(a).


 In addition to the schematic illustration in Fig. 1-(a), Fig. 2 provides experimental justification for (L1) by analyzing unlearning performance across different types of forget data points, categorized based on their response lengths. The motivation is elaborated in Lines 234-239, highlighting that the reference model may exhibit a bias toward generating longer but lower-quality sequences, as noted in (Meng et al., 2024). Consequently, these low-quality, long texts tend to be easier to unlearn. Accordingly, this optimization power allocation could become uneven for unlearning short-response forgotten data points, Indeed, Fig. 2 shows that NPO exhibits a worse unlearning performance, in terms of a greater distance from Retrain, when unlearning the top 50% shortest-length forget data than the longer 50% of the forget set. Analyzing results in Fig. 6 via a mixture of Markov chains also validated the above points.

---

### Author Response · Authors · 2024-11-20
**General Response (GR)**

In the following, we provide several general responses (GRs) to address common questions raised by multiple reviewers. We will also incorporate the corresponding revisions into the revised paper.

---

### Author Response · Authors · 2024-11-24
**Upload the revised paper to the supplementary material.**

We sincerely thank all the reviewers for their valuable feedback on our paper. Following your suggestions, we have made further revisions to the manuscript. To facilitate comparison with the original version, we have uploaded the revised paper to the Supplementary Material, with the main changes highlighted in red. We have added more explanations for technical terms and conducted additional experiments, including divergence and reference model bias analyses. We look forward to your further feedback and suggestions!

---

### Author Response · Authors · 2024-12-02
**Rebuttal summary to all reviewers and ACs**

We sincerely appreciate the valuable feedback from all reviewers and the ACs' management of our submission. During the rebuttal phase, we have made substantial efforts to address the reviewers' problems and improve our submission. Below is a summary of our rebuttal, with the revised manuscript included in the uploaded supplementary material.

**Reviewer x96s:**
1. We added **[new experiments](https://openreview.net/forum?id=Pd3jVGTacT&noteId=Z9cYtRqSgN)** on unlearning vs. data memorization (Table R1) to further [clarify our motivation and the reference model bias](https://openreview.net/forum?id=Pd3jVGTacT&noteId=aouVi1kVTz); Please also see the revised introduction and Sec. 4
2. We elaborated on [the ablation study and analyses about hyperparameters](https://openreview.net/forum?id=Pd3jVGTacT&noteId=0cbuT3Cu9T) that have been presented in the paper.
3. We further clarified our [contributions](https://openreview.net/forum?id=Pd3jVGTacT&noteId=zXCzQUEhLG).

**Reviewer ojtB:**
1. We clarified the concept of [divergence](https://openreview.net/forum?id=Pd3jVGTacT&noteId=1dU2FIB9C9), together with **[new experiments](https://ibb.co/TgcqWSR)** about divergence rate (Figure R1); Please also see the revised introduction in the revised manuscript.
2. We provided further explanation of calling [principled optimization framework](https://openreview.net/forum?id=Pd3jVGTacT&noteId=MDkg0ZMyzF) in the original submission; Please also see the revised introduction.
3. We further clarified [Fig. 1-(a)](https://openreview.net/forum?id=Pd3jVGTacT&noteId=XJKnbKnJRp).
4. We further elaborated on [the novelty of SimNPO](https://openreview.net/forum?id=Pd3jVGTacT&noteId=XAeoL8CtzT).
5. Through **[new experiments](https://openreview.net/forum?id=Pd3jVGTacT&noteId=Z9cYtRqSgN)** on unlearning vs. data memorization (Table R1), we further explained [the reference model bias and provided a detailed analysis of Fig. 3  (a)-(b)](https://openreview.net/forum?id=Pd3jVGTacT&noteId=WDXiDb1VJt). Please also see the revised Sec. 4.
6. We conducted the **[suggested experiments](https://openreview.net/forum?id=Pd3jVGTacT&noteId=FqLRl7AThl)** (Table R2)  to demonstrate the effectiveness of our method on additional models.

**Reviewer HSyZ:**
1. We offered further explanation of [divergence](https://openreview.net/forum?id=Pd3jVGTacT&noteId=ynw7r2EfTF) through **[new experiments](https://ibb.co/TgcqWSR)** about divergence rate (Figure R1). Please also see the revised introduction.
2. We further clarified and expanded our explanation of the [reference model bias](https://openreview.net/forum?id=Pd3jVGTacT&noteId=itX4f1kYnc) and conducted **[new experiments](https://openreview.net/forum?id=Pd3jVGTacT&noteId=Z9cYtRqSgN)** on unlearning vs. data memorization (Table R1) to validate that. Please also see the revised Introduction and Sec. 4.
3. We further elaborated on [the role of $\pi_{\text{ref}}$ in NPO](https://openreview.net/forum?id=Pd3jVGTacT&noteId=iljQ7XpXqY).
4. We further discussed the challenges posed by [short and long samples](https://openreview.net/forum?id=Pd3jVGTacT&noteId=elrRj0Yy49) in NPO with the support of **[new experiments](https://openreview.net/forum?id=Pd3jVGTacT&noteId=EjD8qKrlIY)** (Table R5) and **[new figure](https://ibb.co/sWdxW97)** (Figure R2).
5. We discussed [optimal weight smoothing](https://openreview.net/forum?id=Pd3jVGTacT&noteId=pF2PCONoYC).
6. We further clarified the [contributions](https://openreview.net/forum?id=Pd3jVGTacT&noteId=14SC25oAyO) of our work.
7. According to the [experiments](https://openreview.net/forum?id=Pd3jVGTacT&noteId=4Cn4pgJvFl) already presented in the paper, we discussed and compared our proposal with a simpler weighting strategy.

**Reviewer Fdki:**
1. We further clarified the [contributions](https://openreview.net/forum?id=Pd3jVGTacT&noteId=ntqPWnd6gu) of our work.
2. As validated by the [experiments](https://openreview.net/forum?id=Pd3jVGTacT&noteId=RHW87Plmqw) in the paper, we demonstrated that SimNPO does not introduce additional hyperparameter complexity.
3. We added new discussion and **[new experiments](https://ibb.co/TgcqWSR)** (Figure R1) to explain and validate the [divergence rate of SimNPO](https://openreview.net/forum?id=Pd3jVGTacT&noteId=lStyM4hCNG).
4. We incorporated **[additional mathematical benchmarks](https://openreview.net/forum?id=Pd3jVGTacT&noteId=u8ZFZpIDzA)**, such as GSM8K, to further validate the effectiveness of SimNPO in Table R3 and R4.

We sincerely hope the reviewers recognize the substantial efforts we have made in our rebuttal and paper revisions, and find our responses helpful in addressing their previous comments. We believe these efforts highlight the value of our work and warrant a reconsideration of the ratings to better reflect its contributions. We hope our rebuttal and revision efforts could be acknowledged by all reviewers and thoughtfully considered in the evaluation of our submission.

---

### Meta-Review · Area_Chair_VfKe · 2024-12-18

**Metareview:**

The authors revisited the current unlearning optimization framework, negative preference optimization (NPO), and identified its reference model bias issue, which compromises unlearning effectiveness, particularly for forget data of varying difficulty. To address this, the authors introduced SimNPO, a simple yet effective framework that eliminates reliance on a reference model by leveraging simple preference optimization. The authors provided deep insights into SimNPO’s advantages through both synthetic data analysis and evaluations on existing unlearning benchmarks such as TOFU, MUSE, WMDP, and relearning attacks. For the strengths, the authors revisit the NPO framework and identify its potential weakness–overreliance on the reference model–in LLM unlearning. Building on insights into NPO’s limitations, the authors propose an improved LLM unlearning approach, SimNPO, which extends NPO using a reference-free optimization framework. The authors conduct extensive experiments to showcase the improvements of SimNPO over NPO across various unlearning benchmarks. However, there are several points to be further improved. The paper may somewhat overclaim its contribution. It seems that the main claim of this paper is that current methods might not be effective in handling short strings unlearning, motivating the paper to propose a new method that explicitly pay more attentions to those long strings. The current organisation of the paper is not clear, especially considering the fact that the authors do not discuss what kinds of the weighting mechanism is of our ultimate interest. The authors claim that NPO suffers from over-unlearning. However, comparing the results in Table 3 between NPO and SimNPO for MUSE News, it seems that SimNPO faces the problem of under-unlearning. Some statements and claims are not very clear and sometimes confusing. It would be better if the authors could improve the explanation or discussion for some words or sentences. Moreover, the current presentation is not very good as it is difficult to understand some critical parts of the problem illustration or justification with long-distance cross-reference. For example, there are no related descriptions before or near the position where Fig.1 existed for the first time, and the corresponding caption can not fully explain all the meanings of the figure. Besides, it is unclear about the unique novelty and insights of adopting SimPO to improve NPO in both perspectives of research question and methodology design. The identified issue of NPO is reference model bias, while the reference model bias is not clearly illustrated in Section 4. For example, although we can get that both the forget quality and model utility of SimNPO would be better than NPO, there are only statistics of NPO at Figure 3 (a)-(b). Lacking of comparison makes it hard to intuitively get the meaning of blind allocation and ineffective gradient weight smoothing. Therefore, this paper cannot be accepted at ICLR this time, but the enhanced version is highly encouraged to submit other top-tier venues.

**Additional Comments On Reviewer Discussion:**

Reviewers keep the score after the rebuttal.

---

### Decision · Program_Chairs · 2025-01-22

Reject